

# Abundance of fluorescent biological aerosol particles at temperatures conducive to the formation of mixed-phase and cirrus clouds

Cynthia H. Twohy[1], Gavin R. McMeeking[2], Paul J. DeMott[3], Christina S. McCluskey[3], Thomas C. J. Hill[3], Susannah M. Burrows[4], Gourihar R. Kulkarni[4], Meryem Tanarhte[5], Durga N. Kafle[6], and Darin W. Toohey[7]

[1]Northwest Research Associates, Redmond, WA 98052 USA
[2]Droplet Measurement Technologies, Boulder, CO 80301 USA*
[3]Colorado State University, Fort Collins, CO 80523 USA
[4]Pacific Northwest National Laboratory, Richland, WA 99354 USA
[5]Max Planck Institute for Chemistry, Mainz, 55128 Germany
[6]NASA GSFC, ADNET Systems, Greenbelt, MD 20771 USA
[7]University of Colorado, Boulder, CO 80309 USA
*Now at Handix Scientific, Boulder, Colorado, 80301 USA

*Correspondence to*: Cynthia Twohy (twohy@nwra.com)

**Abstract.** Some types of biological particles are known to nucleate ice at warmer temperatures than mineral dust, with the potential to influence cloud microphysical properties and climate. However, the prevalence of these particle types above the atmospheric boundary layer is not well known. Many types of biological particles fluoresce when exposed to ultraviolet light, and the Wideband Integrated Bioaerosol Sensor takes advantage of this characteristic to perform real-time measurements of fluorescent biological aerosol particles (FBAP). This relatively new instrument was flown on the National Center for Atmospheric Research Gulfstream-V aircraft to measure concentrations of fluorescent biological particles from different potential sources and at various altitudes over the U. S. western plains states in early autumn. Clear-air number concentrations of FBAP larger than 0.8 μm diameter usually decreased with height, and generally were about 10-100 L$^{-1}$ in the continental boundary layer, but were always much lower at temperatures colder than 255K in the free troposphere. At intermediate temperatures where biological ice nucleating particles may influence mixed-phase cloud formation (255K≤T≤270K), concentrations of fluorescent particles were the most variable, and were occasionally near boundary layer concentrations. Predicted vertical distributions of ice nucleating particle concentrations based on FBAP measurements in this temperature regime sometimes reached typical concentrations of primary ice in clouds, but were often much lower. If convection was assumed to lift boundary layer FBAP particles without losses to the free troposphere, better agreement between predicted ice-nucleating particle concentrations and typical ice crystal concentrations was achieved. Ice nucleating particle concentrations were also measured during one flight and showed a decrease with height, and concentrations were consistent with a relationship to FBAP established previously at the forested surface site below. The vertical distributions of FBAP measured on five flights were also compared with those for bacteria, fungal spores and pollen predicted from the EMAC global chemistry-climate model for the same geographic region.



# 1 Introduction

Details of the formation of ice in clouds are poorly understood, especially considering the importance of this phase transition to cloud evolution, climate and the cycling of water and trace constituents in Earth's atmosphere. Water droplets can remain supercooled at temperatures below 273K, and the presence of an ice nucleating particle (INP) reduces the energy barrier required for the phase transformation from liquid to ice. Biological particles have received much interest in the community recently because certain ones tend to nucleate ice efficiently at warmer temperatures than mineral dust particles (Murray et al., 2012). While laboratory studies show mineral dust and some types of biological particles can act as ice nucleating particles (INPs), there is conflicting evidence regarding the importance of biological particles as INPs in the atmosphere (Després et al., 2012). Modeling studies suggest that biological INPs are not very important globally, mainly because their concentrations at cold cloud levels are thought to be relatively low compared to other ice nucleating particles like mineral dust (Hoose et al., 2010).

Primary, or directly-emitted, biological particles are diverse and include bacteria, fungal spores and fragments, viral particles, pollen and plant debris. Number concentrations near the surface are generally ~1-100 L$^{-1}$ for individual types, and can sometimes reach as high as 1000 L$^{-1}$ for all biological particles >0.4 µm in diameter (Jaenicke, 2005). Given the differences in their size and source regions, biological particles have varying lifetimes and are unevenly distributed throughout the atmosphere. Perring et al. (2015) measured a wide range of fluorescent (potentially biological) particle types at low altitudes across the southern United States and found that they comprised about one-fourth of the number concentration of particles larger than 1 µm in diameter. However, there are few in-situ measurements of biological particle concentrations at cloud levels. Fulton (1966) found that concentrations of microorganisms usually declined with height up to the limit of their measurements, about 3 km, in the southern United States. DeLeon-Rodriguez et al. (2013) reported that in air influenced by Atlantic hurricanes, about 20% of submicron particles were biological. Pratt et al. (2009) and Creamean et al. (2013) reported that biological particles sometimes dominated ice residuals in mid-level clouds over the western United States. However, Cziczo et al. (2013) found biological particles were essentially absent in cirrus anvil outflow from various regions. In the latter study, cirrus crystal residual nuclei were found to be primarily mineral dust and industrial metals. Twohy (2014) also found that anvil cirrus from Atlantic tropical storms contained mostly mineral dust and industrial metals.

While mineral dust mass is dominated by supermicron particles, dust number concentration is dominated by submicron particles (Weinzierl et al., 2009; Chen et al., 2011). Thus, atmospheric motions can sometimes bring them to the upper troposphere where cirrus clouds form. Primary biological particles tend to be less numerous in the atmosphere and are usually supermicron in size (Schneider et al., 2011; Després et al., 2012), and so are expected to be present in relatively low quantities at high altitudes. However, biological particles may be critical to generating first ice at intermediate altitudes in mixed-phase clouds, since mineral dust becomes a less efficient ice nucleator at temperatures warmer than about 255K (Murray et al., 2012).

Yet what are the concentrations and types of biological particles as a function of height, or temperature, in the atmosphere? Aerosol particles are ubiquitous in the planetary boundary layer, because they usually have surface sources, and the stable boundary layer cap tends to keep them there. However, convective air motions can



sometimes lift even large particles above the boundary layer, as evidenced by Asian or Saharan dust storms. In addition to their ice nucleating properties, both dust and biological particles are known to act as cloud condensation

nuclei due to their relatively large sizes and their ability to absorb or adsorb water (Möhler et al., 2007; Twohy et al., 2009; Kumar et al., 2011). Thus their abundance at higher altitudes will be influenced not only by their source strength, vertical lifting, and sedimentation rate, but also by their rate of removal in precipitating cloud systems.

Kafle and Coulter (2013) presented seasonally averaged aerosol optical properties at various Atmospheric Radiation Measurement (ARM) program sites based on micropulse aerosol lidar data. They found that while most

aerosol particles were confined to the boundary layer, some particles were detected above it up to the extent of their analysis (4 km). To evaluate the tendency for particles to escape the boundary layer over the United States Great Plains states, we analyzed data for the Southern Great Plains site in Oklahoma up to 10 km altitude. Fig 1a shows aerosol extinction as a function of height for each season, averaged over non-cloudy days for the 2007-2010 period. Aerosol particles are present up to 5-6 km in most seasons, well above the typical daytime boundary layer depth of 2

km or less (Kafle and Coulter, 2013). Fig. 1b shows the standard deviation of the autumn aerosol profile as a function of height. It indicates that there is substantial variability from day to day in the aerosol loading, with about a third of the extinction profiles substantially above or below the mean profiles. While aerosol extinction is usually dominated by sub-micron particles, larger particles generated from local sources or long-range transport (Mishra et al., 2013) also may reach these mid-tropospheric heights.

Fig. 1c shows the mean 24-hour temperature profiles corresponding to the same time period as the ARM-site aerosol profiles in Fig. 1a and 1b. The profiles demonstrate that in this location, mean boundary layer temperatures are always warmer than freezing, so ice is unlikely to form in clouds near the surface. At 2 km, mean temperatures range from 272K-287K, depending on season, while at 5 km, temperatures drop to 255K-267K. The 2-5 km altitude range is of critical importance in this region, because this is not only where aerosol particles are

relatively abundant in the free troposphere, but where temperatures may be cold enough for biological particles to nucleate ice (for example, Murray et al. (2012)). Fluorescent biological particle measurements described later in this paper were taken near Boulder, Colorado, where seasonally-averaged surface temperatures are about 2K-5K colder than at the ARM site.

Determining the importance of biological particles to atmospheric ice phase transitions is difficult due to

the limited data on lofted biological material at altitudes where temperatures are below 273K. An additional source of uncertainty for the role of biological particles in clouds is their huge range of ice nucleating characteristics, with number fractions for ice nucleating types ranging between about $10^{-7}$ at 270K to as high as 1 at 250K for certain ice nucleating bacteria (Després et al., 2012). The situation is additionally complicated by indications that ice-nucleating material of biological origin also can be present in submicron particles that could be more easily be lofted in the

atmosphere. Based on filtration of melted hailstones and decaying leaf litter, substantial ice nucleating material was found to be smaller than 100 nm in size (Vali, 1966; Schnell and Vali, 1973). More recently, O'Sullivan et al. (2015) note the presence of unidentified "nano-INP" (well below 100 nm) of biological origin associated with fungi and pollen and present in soils. Wilson et al. (2015) identified ice nucleating material from the sea surface microlayer that passed through a 200 nm filtration system, found similar INP in diatom exudates, and proposed phytoplankton





exudates as INP sources. (It is not clear if these ice nucleating entities would display fluorescence, a characteristic commonly used to indicate primary biological origin.) Despite these indications that small particles might be involved in ice nucleation, a recent paper by Mason et al. (2015) showed that the majority of INP in the atmosphere at six different North American sites were supermicron in size.

Biological particles are difficult to measure in-situ, especially at the lower concentrations present far from surface sources. A new instrument based on fluorescence, the Wideband Integrated Bioaerosol Sensor (WIBS-4A), was used to measure biological aerosol particles in real time on an aircraft in September and October 2013. Particles larger than 0.8 µm (both fluorescent and non-fluorescent) were measured at temperatures of about 280K to 220K, ranging from the boundary layer to the upper troposphere, over the United States Great Plains region. Flights were mostly over undeveloped grassland or cropland toward the end of the growing season, with one flight over a forested

site. To our knowledge, these are the first vertical profiles of fluorescent biological aerosol particles (FBAP) presented over a wide range of temperatures where ice is known to form. Filter samples for determining INP number concentration versus temperature spectra were also collected on the aircraft and at a heritage ground-based site. Vertical distribution of biological aerosol particles in the same region derived from a global chemistry-climate model were compared to the aircraft measurements of FBAP to test model predictions. Finally, concentrations of ice

nucleating particles estimated from WIBS measurements were compared to typical concentrations of ice crystals at mixed-phase cloud temperatures. While this technique has substantial uncertainty, since there may be biogenic INPs that do not fluoresce or are not large enough to be detected by the WIBS-4A, it provides a reasonable first step for quantifying the potential influence of FBAP on ice nucleation in this portion of the atmosphere.

## 2 Methods

### 2.1 Experiment

Particles were observed using the U. S. National Science Foundation's Gulfstream-V aircraft during the *Instrument Development and Education in Airborne Science* (IDEAS) 2013 field program. The goals of the program were to improve the capability of instrumentation for future National Science Foundation airborne deployments and to provide opportunities for students to learn about observational science. Data from five flights between 26 September

2013 and 21 October 2013 were used to examine fluorescent biological particle concentrations as a function of height in clear air. Fig. 2 shows the location of the sampling during the flights, superimposed on a 2010 land use map from the United States Geological Survey. The aircraft was based near Denver, Colorado and the flight tracks were usually to the north and east over rural Wyoming, Nebraska and South Dakota. One flight (8 October) was over the forested "BEACHON" project site southwest of Denver (Ortega et al., 2014) that included measurements

focused on ice nucleating properties of biological particles (Huffman et al., 2013; Tobo et al., 2013). Remaining flights were over arid areas and cropland, and downwind of forest, shrubland, cropland, and some urban areas. In addition to the real-time WIBS measurements, filter samples were collected on several flights and at the ground-based forested site while the aircraft flew overhead. These samples were later subjected to extraction and analysis for ice nucleating particles using an immersion freezing method (Hill et al., 2014).





## 2.2 Aircraft sampling

To enhance the concentrations of biological particles, a National Center for Atmospheric Research counterflow virtual impactor (CVI) with the counterflow turned off was used as a subisokinetic inlet (Krämer and Afchine, 2004). The CVI inlet was made of titanium and was mounted on the bottom of the G-V aircraft where the airflow is less disturbed by the shape of the aircraft itself (King, 1984), with its entry tip 6.4 m behind the aircraft nose and 0.28 m away from the fuselage skin. Under subisokinetic conditions, the smallest particles follow streamlines when entering the inlet, while larger particles deviate from the streamlines and are concentrated inside the inlet. Computational fluid dynamics modeling was used to calculate aspiration efficiency and transmission efficiency of the subisokinetic inlet as a function of particle size for two different aircraft speed cases, simulating low and high altitude sampling. A detailed aerosol transport model (von der Weiden et al., 2009) was then used to calculate transmission efficiency in the plumbing downstream of the inlet itself. The net inlet efficiency (aspiration and transmission) as a function of particle size and airspeed was then applied to the particle measurements to correct them back to ambient atmospheric conditions, as presented below. Further details of these calculations and assumptions therein are presented in Appendix A.

## 2.3 Fluorescent biological particle measurements

The Droplet Measurement Technologies Wideband Integrated Bioaerosol Sensor (WIBS-4A) was used to perform real-time measurements of fluorescent biological aerosol particles (FBAP) from the aircraft. Most biological particles contain amino acids and other compounds that fluoresce at wavelengths detected by WIBS technology, and most non-biological particles fluoresce more weakly or at different wavelengths. Therefore the WIBS-4A may be used to distinguish biological from non-biological particles (Pöhlker et al., 2012; Huffman et al., 2013). The WIBS measured fluorescent emissions on a single particle basis for two ultraviolet excitation wavelengths and in two emission windows (Kaye et al., 2005; Perring et al., 2015). Particles were first sized using elastically scattered light from a 635 nm diode laser, which also served as a trigger for the fluorescence measurements. Two filtered Xenon flashlamps (Xe1 = 280 nm, Xe2 = 370 nm) were then fired in sequence and any resulting fluorescence was monitored by two photomultiplier tubes filtered to measure light between 310-400 nm (FL1) and between 420-650 nm (FL2). Flashlamp timings were optimized using 2 μm fluorescent polystyrene latex spheres (Thermo Scientific). The Xenon flashlamps were limited to an upper rate of approximately 125 Hz due to recharging requirements, so final particle concentrations were corrected for any particles missed during recharge periods.

The combination of two excitation wavelengths and two emission windows provides three useful channels of fluorescence information (Perring et al., 2015). Channel A (previously referred to as FL1_280) is defined as 280 nm excitation and 310-400 nm emission; Channel B (FL2_280) is defined as 280 nm excitation and 420-650 nm emission, and Channel C (FL2_370) is defined as 370 nm excitation and 420-650 nm emission. Particles were classified as fluorescent if they emitted light in a single channel or combination of channels above background levels. Background fluorescence signals were determined from "forced trigger" periods where the Xenon flashlamps were fired in the absence of particles in the chamber. We calculated the raw fluorescence signal average and



standard deviation in each channel and assigned a fluorescence threshold equal to the mean plus three standard deviations.

    Seven possible combinations of fluorescent emission response are possible (A only, B only, C only, AB but not C, AC but not B, BC but not A, and ABC. The use and understanding of these channel combinations to define biological particle types is still maturing within the WIBS user community. Fluorescence in both Channels A and C

correlates with the presence of both the amino acid tryptophan and the coenzyme NADH (nicotinamide adenine dinucleotide), likely indicating actively metabolizing organisms such as bacterial cells (Pöhlker et al., 2012). Therefore, in previous work focusing on interpretation of WIBS fluorescent signals, the AC and ABC channels have been combined into a category sometimes called FBAP or FL13 (Gabey et al., 2011). This has been considered to be a conservative estimate for fluorescent biological particles, though other interpretations of the signals have also been

applied (e.g., Wright et al. (2014)). More recently, Perring et al. (2015) used all categories of fluorescent particles to present a more inclusive interpretation of the WIBS data. In the Results section, we present two different FBAP concentrations ("low" and "high") to represent the uncertainty currently inherent in measuring biological particles via this fluorescent method.  The lower bound or conservative estimate encompasses only particles that fluoresce in *both* channels A and C (formerly FL13), and facilitates comparisons with many earlier measurements. The high, or

expected upper bound of FBAP presented here includes all categories of particles fluorescing in Channels A *or* C. We preclude particles fluorescing in Channel B only, since this channel may be influenced by anthropogenic, non-biological particles (Gabey et al., 2011; Toprak and Schnaiter, 2013), and background fluorescence signal for this channel was higher than usual in the particular instrument used.

    The WIBS-4A deployed for IDEAS was modified to provide better flow control and measurement for

operation behind the CVI. The total flow was regulated by an Alicat mass flow controller operating in volumetric mode. The total volumetric sample flow was converted to a 0-10 VDC signal and passed to the CVI data system to provide real-time control based on instrument flows. The WIBS-4A was located near the middle of a standard 1.27 m high G-V rack and connected to the sampling inlet with stainless steel and conductive silicon tubing. Sizing calibration was verified before the project using 2 μm PSLs, a size that is representative of the peak size (2-3 μm

diameter) for the measured fluorescent particles. However, calibration may deviate by as much as 20% for larger sizes, and variation in scattering with size can be +/- 15% (A. Perring, personal communication). In addition, biological particles may have different shapes and refractive indices from polystyrene spheres. Therefore, net uncertainty in sizing of biological particles via the WIBS-4A may be as high as 50%. This corresponds to about a 40% error in the size-dependent concentration corrections for inlet aspiration and transmission efficiency described

in Appendix A. The 1Hz clear-air WIBS data were averaged over 200 s of flight time to reduce uncertainty at low particle concentrations. With the subisokinetic enhancement factor, this corresponds to about 5 liters of particles collected during each sampling period. WIBS concentrations are reported in ambient (not standard) $L^{-1}$, for consistency with accepted reporting of ice crystal number concentrations in the cloud physics literature.

    Error bars on the number concentration plots represent the root-sum squared (RSS) uncertainty from three main

sources: 1) counting statistics, 2) WIBS concentration errors, and 3) uncertainty in inlet efficiency calculations. For the first, we use $N^{1/2}$ for positive counts and $(N+1)^{1/2}$ for zero counts (FDA et al., 2004). For the second, we estimate



20% due primarily to forced-trigger baseline uncertainty. The RSS uncertainty of the calculated inlet efficiency is 53% (incorporates 40% due to the WIBS sizing uncertainty given above and 35% estimated uncertainty in model efficiency calculations). Propagated concentration uncertainty varies with concentration magnitude, but is less than 60% in most cases. This is usually much smaller than the difference in concentrations resulting from the conservative and liberal approaches in defining FBAP, as described earlier.

**2.4 Ice nucleating particle (INP) filter samples**

Particles were collected onto filters, and then re-suspended in water for measurement of INPs using the immersion freezing method. INP measurements used the Colorado State University ice spectrometer (Hill et al., 2014; Hiranuma et al., 2015a), a device in which an array of liquid aliquots in a temperature-controlled block can be monitored for freezing events as temperature is decreased.

Filters used were 0.2 μm-pore-diameter, 47 mm diameter Nuclepore™ track-etched polycarbonate membranes (Whatman, GE Healthcare Life Sciences). Filters were cleaned before sampling by immersion in 15% $H_2O_2$ for 10 min, followed by two rinses in deionized water (18 MΩ and 0.2 μm-diameter-pore filtered) and one rinse in deionized water that had been filtered through a 0.02 μm-pore-diameter syringe filter (Anotop, Whatman, GE Healthcare Life Sciences), then dried on foil and loaded into filter units with sterile tweezers. All preparations were performed in a laminar flow cabinet (<0.01 particles mL$^{-1}$). On the G-V aircraft, 47 mm in-line aluminum filter housings (Pall Corporation) were used to contain sampling filters. These units were cleaned before use by disassembly, immersion in 10% $H_2O_2$ for 30 min followed by three rinses in deionized water (18 MΩ and 0.2 μm-diameter-pore filtered), and then dried by removal of excess water and placement on foil in a laminar flow cabinet. Aircraft filters were operated at 5 L min$^{-1}$ through a 0.48 cm inner diameter stainless steel line that connected to the CVI. Collection onto the surface or into the pores of the Nuclepore filters should have exceeded 90% for all particle sizes at the flow rates used on the basis of filter specification and theoretical collection efficiencies (Spurny and Lodge, 1972). After particle collection, filters were stored frozen in sealed sterile petri dishes until processed.

During ground-based sampling at the BEACHON site on 8 October, open-faced Nalgene sterile filter units (Thermo Fisher Scientific Inc.) were pre-loaded with the same type of Nuclepore filters as used on the aircraft. Sampling was conducted over a period (11:34 to ~14:30 MST) that encompassed the time of the aircraft overpasses (12:50 to 13:25 MST). One filter unit was placed 14 m above ground and sampled with a flow rate of 8.5 L min$^{-1}$, while the other was placed 1 m above ground, sampling at 9.0 L min$^{-1}$. Units were returned intact on ice and stored at -20°C until processed.

For processing, filters were transferred to sterile, 50 mL Falcon polypropylene tubes (Corning Life Sciences), immersed in 5.0 mL of 0.02 μm-pore-diameter-filtered deionized water, and tumbled for 30 min at 60 cycles min$^{-1}$ in a rotator (Roto-Torque, Cole-Palmer) to re-suspend particles. Measurements of immersion freezing by the re-suspended particles were made on this suspension and 15-fold dilutions of it to extend measurements to lower temperatures. Liquid suspensions were distributed into 32 aliquots of volume 80 μL in 96-well PCR trays (μCycler, Life Science Products) which were then capped with polystyrene lids (Nunc





microwell plates, Thermo Fisher Scientific Inc.) and transferred to the ice spectrometer. The numbers of wells frozen were counted at 0.5 or 1°C intervals during cooling at a rate of -0.3°C min$^{-1}$, and cumulative numbers of

INPs per volume of liquid as a function of temperature were estimated using the formula, -ln $f_u(T)/V$, where $f_u$ (T) is the unfrozen fraction at a given temperature and V is an aliquot volume (Vali, 1971). This formula accounts for the fact that each aliquot may hold more INPs than the first one that freezes. Correction for any frozen aliquots in the water used for suspension was made in all cases. Uncertainties are given as binomial sampling confidence intervals (95%) (Agresti and Coull, 1998). Conversion to INP number concentrations in

ambient L$^{-1}$ was made using the sample volumes and correcting for the inlet aspiration and transmission efficiency discussed in the Appendix for the aircraft samples.

Two blank filters were also collected during aircraft flights and analyzed to constrain the influence of possible contamination during sampling. This was necessitated by the fact that the sample volumes for aircraft filters were much smaller than the ~2,000 L collected by ground-based filters. Since INPs released

from the blank filters differed, corrections for and tests of significance between sample and blank INPs at each temperature were performed separately for each blank. Tests of significance between sample and blank used Fisher's Exact Test (Fisher, 1922) to derive exact *p* values for the likelihood of the difference in proportions of wells unfrozen (i.e., not containing an INP) between sample and blank at each temperature.

The *p* value is given by: $p = \frac{(a+b)!(c+d)!(a+c)!(b+d)!}{a!b!c!d!n!}$

where *a* and *b* are the numbers of wells unfrozen and frozen, respectively, in the sample, and *c* and *d* the same for the blank, at each temperature. *n* is the combined total number of aliquots being tested in both samples.

### 2.5 Global chemistry-climate model

The global chemistry-climate model ECHAM5/MESSy-Atmospheric Chemistry (EMAC) (ECHAM version 5.3.01, MESSy version 1.9; Jöckel et al. (2005)) was used to simulate the emissions and transport of biological particles. Model simulations were conducted in T63L31 resolution (i.e., 210 km x 210 km at the equator, with 31 vertical levels up to a model top of 10hPa). The model dynamic scheme was weakly nudged (Jeuken et al., 1996; Jöckel et al., 2006; Pozzer et al., 2012) towards the analysis data of the European Centre for Medium-Range Weather

Forecasts (ECMWF) operational model (up to 100 hPa), such that the meteorology in the model results shown here is consistent with the time period during which the campaign took place. The model simulation was initialized for 1 January 2012, to allow ample spin-up time. Simulation results were used from the times of the aircraft flights in September and October of 2013.

Bacteria emissions were calculated using the best-estimate number fluxes from Burrows et al. (2009b) with

the minor modification that the flux from land ice was set to zero. Fungal spore emissions were calculated following Heald and Spracklen (2009) and pollen emissions were calculated following Jacobson and Streets (2009). The following geometric mean diameters (d) were assumed for the different particle classes. For bacteria, d was set to



4μm (continental sources) or 2.4 μm (marine sources), following values reported for the count-median-diameter of bacteria-carrying particles, which may include bacteria borne on larger particles such as dust and leaf litter, and/or

clumps of bacteria (Shaffer and Lighthart, 1997; Tong, 1999; Tong and Lighthart, 2000; Wang et al., 2007). For fungal spores, d=4 μm (Hussein et al., 2013) was used, and for pollen, d=20 μm (Niklas, 1985; Di-Giovanni et al., 1995).

All particle classes were treated as having a lognormal distribution with modal scale parameter σ=1 and with a density of 1 g cm$^{-3}$. Further, all particles were treated as CCN-active when calculating particle removal

processes, as described in Burrows et al. (2009a). The sensitivity of particle transport and removal processes to these and other model parameters has been characterized in detail for an earlier version of the EMAC model (Burrows et al., 2013). All biological particles were transported as passive tracers, i.e., their concentrations were influenced by model processes (such as precipitation scavenging), but bioaerosols did not interact with radiation or influence cloud microphysical properties. The sedimentation and dry deposition of the particles are treated as described in Kerkweg

et al. (2006). The wet deposition of the particles is described in Tost et al. (2006).

## 3. Results

### 3.1 Comparison with ground and tower data at forested site

A previous ground-based study suggested that the population of ice nucleating particles at the forested BEACHON site near Manitou Springs, Colorado was dominated by biological particles (Tobo et al., 2013). Prenni et al. (2013)

and Crawford et al. (2014) found enhanced bioaerosol concentrations after rain events in this region, and Prenni et al. (2013) showed that they were correlated with concentrations of ice nucleating particles as measured by a continuous flow diffusion chamber (CFDC).

During IDEAS, the aircraft flew over the same site on 8 October 2013. First we present fluorescent biological particle concentrations and INP concentrations from filter measurements taken on the aircraft over the

BEACHON site and compare them with similar measurements taken simultaneously on the ground or at the canopy top at the same site. The aircraft spiraled down over the ponderosa pine site from 3638 m to 897 m above ground level (AGL), near midday. Fig. 3a shows that about 10-60 L$^{-1}$ particles at diameters larger than 0.8 μm appeared to be biological above the forest canopy, with a sharp decline above the top of the temperature inversion at about 1.7 km. WIBS-4A FBAP measurements at the lower altitudes are similar to those measured earlier by another

fluorescent-based instrument, an ultraviolet aerodynamic particle sizer (UV-APS, TSI manufacturer) at the same site (Schumacher et al., 2013). Using the UV-APS, Schumacher et al. (2013) measured mean FBAP concentrations (>1.0 μm diameter) of about 30 L$^{-1}$ in summer and 17 L$^{-1}$ in fall of 2011 at 4 m AGL.

Fig. 3a also shows the atmospheric temperatures over which filters were collected on the aircraft for offline INP spectral analysis. One filter (6A) was collected over a range of altitudes as the aircraft spiraled down over the

ground site, and another (6B) was taken while flying level at 1067 m AGL in a racetrack pattern over the site, within the atmospheric boundary layer. Sample volumes were ~37 L for 6B and ~142 L for 6A. Fig. 3b shows INP temperature spectra from the off-line filter analysis. As discussed above, two near-surface filters were collected



simultaneously with aircraft filter samples 6A and 6B: one at 1 m above the ground and one in a tower at 14 m. INP spectra determined from the 1 m and 14 m filters (green triangle points) are very similar to each other. The INP

spectra for the aircraft filters are shown as upper bound (open points) and lower bound (filled points) data, in blue and red, respectively, for samples 6A and 6B in Fig. 3b. The bounding values are based on subtracting the "background" or contamination INP numbers of the blank filters from the observed INP numbers per filter. Upper INP bounds were derived by subtraction of the cleaner of the two blanks and lower bounds by subtraction of the more contaminated blank. Upper and (positive) lower-bound data are connected with vertical lines to give a sense of

the range of likely INP number concentrations from the collections made at altitude. Data points fulfilling a Fisher's exact test for significance (Sect. 2.4) between samples and background are plotted as the largest points; these points were all upper-bound points. These data present a large uncertainty, but indicate that INP number concentrations at 1067 m (6B) were about the same to about a factor of four lower than at the forest canopy top (14 m). However, the sample collected over a range of altitudes, primarily in the free troposphere, exhibited much lower INP

concentrations than the boundary layer filter, by a factor of approximately five. Lower INP number concentrations aloft occur in association with decreasing FBAP concentrations with height measured at the BEACHON site (Fig. 3a), and are expected given the probable canopy source of INP at this forested site (Crawford et al., 2014).

In Fig. 3c, INP filter-based spectra from the three lowest altitudes are superimposed on in-situ INP results

obtained previously (Tobo et al., 2013) at 1 m using the CSU continuous flow diffusion chamber (light grey diamonds). The new measurements from IDEAS are quite consistent with the range of number concentrations observed using the CFDC during summer 2011. Additionally, ice nucleating particle concentrations were estimated as a function of WIBS FBAP concentrations measured from the aircraft, using a recent parameterization by Tobo et al. (2013) based on the concentration of FBAP >0.5 µm. Using measured low-level FBAP concentrations 10 L$^{-1}$ to

60 L$^{-1}$ (approximate low and high values in Fig. 3a), INP concentrations estimated from the WIBS data are shown as the two dashed black lines in Fig. 3c. The predicted INP number concentrations bracket the BEACHON and IDEAS data well, with all of the observed INP number concentrations falling within the estimated values.

It is important to note that while the Tobo et al. (2013) study showed that FBAP are correlated with INPs at this site, other particle types, such as soil dust, may still be important contributors to INP number concentrations in

the region, particularly at lower temperatures and for drier conditions (Prenni et al., 2013). Also, even if all the INP activity was contributed by biological particles, only a relatively small percentage of them would be expected to nucleate ice at mixed-phase cloud temperatures. For example, Fig. 3c shows that for a moderate FBAP value of ~30 L$^{-1}$, only about 0.01 L$^{-1}$, or 0.03%, of these particles would be expected to produce ice at 263K and about 0.3 L$^{-1}$, or 1%, at 253K. Additionally, these numbers are based on boundary layer measurements and only become relevant if

ice nucleating particles actually reach regions of the atmosphere with humidities and temperatures conducive to forming clouds. On this particular day, biological particles were present at low concentrations (<1 L$^{-1}$) at temperatures below about 272K, where mixed-phase clouds may form (Fig. 3a). Next, we explore the variation of FBAP with temperature measured on five different flights in the region, and compare results with those of a global model. In Sect. 4, implications for ice formation at mixed-phase cloud temperatures are discussed.





### 3.2 Vertical distribution of clear-air FBAP for five flights


Fig. 4 shows the distribution of FBAP measured as a function of ambient temperature on all five flights over Colorado, Wyoming, Kansas and Nebraska. All flights took place during mid-day hours, when the convective boundary layer is expected to be near its maximum (Nilsson et al., 2001), with its top typically at temperatures warmer than about 275K. Clear-air profiles show a general decrease of fluorescent biological particle concentration

with decreasing temperature. Since the WIBS instrument measures particles with a range of fluorescent characteristics, the expected lower bounds on FBAP (only particles that fluoresce in both Channels A and C) are shown with green circles, and the expected upper bounds (particles with broader fluorescent characteristics as described in Sect. 2.3) are shown in magenta circles for each sampling period. These two values can differ by up to an order of magnitude at each location, indicating that more characterization studies of particle type vs. WIBS

response would be very valuable. Even given the uncertainty in what should be characterized as biological, important conclusions can be made. First, FBAP were typically ~10-100 L$^{-1}$ at warm temperatures near the surface, but much lower, between 0-3 L$^{-1}$ at cold, cirrus cloud temperatures. For many mid to high altitude areas, the atmosphere was essentially devoid of fluorescent biological particles. However, two flights are of special interest. On 1 October, a wide range of biological particle concentrations was observed when the aircraft flew a 60-km box

pattern at a constant altitude (~270K) for 90 minutes, indicating that strong variations in FBAP can occur in relatively small areas. On 16 October, relatively high FBAP concentrations (up to 30 L$^{-1}$ for upper-bound values) were observed at temperatures as cold as ~255K.

Fig. 5 shows the clear-air data for all flights plotted together on a linear scale to highlight regions with higher FBAP concentrations. The lower bound estimates of which fluorescing particles >0.8 µm are biological are

given in the plot on the left (Fig. 5a), and the expected upper bounds on the right (Fig. 5b). The approximate homogeneous freezing region, heterogeneous nucleation region and the warmer portion of the latter where microbial INP are expected to be most important (Murray et al., 2012) are shaded in grey, blue and green, respectively. FBAP particle concentrations were highly variable at any given temperature, but particularly in the temperature region where they are likely most important to ice formation in mixed-phase clouds. These temperatures occurred above the

atmospheric boundary layer, well above the surface where biological particles are generated. For three of the five flights, concentrations were much lower in this region of the atmosphere than in the boundary layer. However, for two flights (10/1/13: purple, and 10/16/13: red), tens per liter of potential biological particles sometimes reached higher altitudes and colder temperatures where ice might form heterogeneously. On these days, the presence of higher numbers of biological particles suggest that the ones active as ice nucleating particles might be able to

influence mixed-phase cloud development. This possibility is explored more quantitatively in the next section.

The WIBS-4A data can also determine size of particles both fluorescent and non-fluorescent (based on scattering of 635 nm light). Most of the measured fluorescent particles were relatively large, with a mode size of ~2-3 µm, consistent with the studies at the BEACHON forested site (Huffman et al., 2013). Fig. 6 shows a histogram of the ratio of fluorescent particle concentration to total particle concentration (larger than 0.8 µm) using data from all

five flights. The mean percentage of all large particles that fluoresced in the two WIBS-4A channels was about 3% for the conservative definition and 11% for the more liberal definition of which fluorescent particles are biological.



This is also consistent with other studies, for example, Tobo et al. (2013).

### 3.3 Comparison with global chemistry-climate model

The EMAC global atmospheric model simulates emissions, transport and removal of primary biological aerosol
particles (fungal spores, bacteria, and pollen). Figure 7 compares the modeled total concentrations of these three
particle types with FBAP concentrations observed by the aircraft for the same five days, with black diamonds
interpolated from the aircraft measurement altitude, location and time. To exhibit modeled spatial variation and
detailed vertical profiles, orange diamonds show modeled concentrations for all 25 grid boxes encompassing the
entire IDEAS sampling domain at 1800 UTC for each day. In general, the model simulates concentrations in the
boundary layer that are within the range of what is typically observed, and the decrease in concentration with
increasing altitude qualitatively agrees with the FBAP observations. However, the model frequently underpredicts
observed concentrations in the free troposphere, and observed concentrations often decline with height more rapidly
than the model predicts. It is also interesting to note that the model predicts a large variability in concentrations at
the same altitude throughout the region (orange diamonds).

410            Possible explanations for discrepancies between the model and observations include underprediction of
bioaerosol sources including via long-range transport, overprediction of the rate of removal by dry and wet
deposition of particles, and underprediction of turbulent exchange between the boundary layer and the free
troposphere. It is also possible that important contributions to the observed FBAP may not be represented in the
model. For instance, the model does not include representations of leaf litter or arable soil emissions, which may
contribute to the observed FBAP. Also, the most important source of primary biological aerosol particles in the
model is bacteria, with emissions that are constant in time, representing inferred "background" emissions. In reality,
bacteria emissions may exhibit seasonal or diurnal cycles, and may also be substantially higher in agricultural
regions during periods of harvesting and other agricultural activity. Because these flights took place during
September and October, harvesting of crops such as corn in the study area could plausibly have increased emissions
of bacteria and other primary biological aerosol particles to as much as an order of magnitude more than typical
background amounts (Lindemann et al., 1982; Lighthart, 1984; Lindemann and Upper, 1985).

### 4 Implications for ice nucleation

The WIBS-4A data indicate that large biological particles do sometimes reach altitudes and temperatures
characteristic for heterogeneously nucleated ice. Yet, since only a small subset of all biological particles nucleates
ice at these temperatures, how important may the measured concentrations of FBAP be to ice formation under these
conditions?

Ice crystal concentrations measured in deep frontal clouds during IDEAS were as high as 80 $L^{-1}$, using a
2D-C probe with anti-shatter probe tips (Korolev et al., 2013). However, these frontal cases were likely impacted by
ice nucleated through homogeneous freezing above, sedimenting into warmer temperatures below. In addition,
under some conditions ice crystals may be formed by secondary processes (not involving aerosol particles; e.g.,




Hallett and Mossop (1974)). Cooper (1986) compiled data from several experiments, including some in the same region as this study, where ice was assumed to occur only from primary heterogeneous ice nucleation. He found that the number concentration of primary ice varied with temperature, and could range by a factor of 100 even for similar temperature and cloud conditions. Nevertheless, 68% of the measurements from different parts of the world and in

different cloud types were within about a factor of six for a given temperature between about 248K to 268K. The best fit relationship was $\log_{10}(N_{ice})=-2.35-0.135T_C$, where $N_{ice}$ is in $L^{-1}$ and $T_C$ is temperature in °C. Using the Cooper (1986) relationship, one would expect about 10 $L^{-1}$ of primary ice at 248K, but much lower concentrations, <0.1 $L^{-1}$ at temperatures warmer than 263K. In the Colorado/Wyoming region, Twohy et al. (2010) examined ice concentrations in orographic wave clouds without upper-level seeding clouds, where ice concentrations were

presumed to be associated with primary nucleation from upstream particles. At temperatures of 244K-249K, ice concentrations were typically 1-5 $L^{-1}$, with a maximum of 17 $L^{-1}$ in one case. These values are similar to those obtained with the Cooper (1986) parameterization for the same temperature range.

To compare these anticipated concentrations of primary ice formed through heterogeneous nucleation with potential INP concentrations in the IDEAS sampling region, INP number concentrations were estimated from WIBS

FBAP concentrations using Eqn. 3 of Tobo et al. (2013) (T2013). This parameterization was based on INP concentrations measured using a continuous-flow diffusion chamber and FBAP concentrations measured by the UV-APS at the BEACHON forested ground site in July-August 2011. The parameterization was shown to agree well with IDEAS field data for the BEACHON location, as discussed in Sect. 3.1. The UV-APS measures FBAP >0.5 μm diameter and the WIBS >0.8 μm diameter, but this is a minor difference since Schumacher et al. (2013) found

that nearly all FBAP measured by the UV-APS at the BEACHON site were larger than 1 μm in size. Our analysis was conducted for the temperature range of the BEACHON INP data set for which it was derived (243K-263K), plus an extrapolation to seven degrees warmer to incorporate a broad range of temperatures where biological INPs are potentially important.

Predicted INP number concentrations as a function of ambient temperature are shown in colored circles

(T2013) in Fig. 8a. Points extrapolated outside the parameterization's data range are colored grey (T2013E) to indicate that they have no basis in existing measurements. Predicted INP concentrations are highly variable, as expected given the intra- and inter-flight variability of the FBAP concentrations in this temperature range. Even with the most liberal interpretation of which fluorescent particles are biological (magenta points) predicted INP concentrations are <0.5 $L^{-1}$. The Cooper (1986) primary ice concentrations (C1986) are plotted as blue hexagons for

reference against the INP concentration predictions. Most of the predicted INP concentrations from fluorescent biological particles measured by the WIBS-4A are well below expected concentrations of primary ice in clouds. However, predicted INP concentrations for the highest FBAP concentrations at temperatures ~268K and 256K are within a factor of two of typical ice concentrations and those at ~268K sometimes exceed them.

Also shown for reference are predicted INP number concentrations from the parameterization of DeMott et

al. (2010), D2010. Brown squares marked are within the temperature range of the original measurements and lighter colored squares are extrapolated to warmer temperatures. The DeMott et al. parameterization was developed to relate INP to the number concentration of all aerosol particles > 0.5 μm diameter, and includes data from the IDEAS



sampling region. WIBS particle concentration >0.8 μm (both fluorescent and non-fluorescent) was substituted as a proxy for >0.5 μm particle concentration measured in DeMott et al. (2010). This size difference will underestimate
concentrations by a factor that depends on the aerosol distribution, but could be expected to be at least a factor of two based on available data from past projects. This parameterization can be expected to reflect potential INP concentrations for the aerosol scenario present at the time of IDEAS measurements, whereas the Cooper (1986) parameterization reflects average conditions of observed ice concentrations over potentially different aerosol scenarios. All these parameterizations suggest variable contributions of biological particle influences on total INP
and cloud ice concentrations at different altitudes and at different times.

The FBAP data presented here represent clear-air conditions, where transport from the boundary layer may not be very active. Under conditions of widespread uplift or strong convection that induce condensation of water vapor, higher concentrations of FBAP might be lifted into the free troposphere, as suggested by Wright et al. (2014). To assess whether FBAP measured in the boundary layer during IDEAS would be sufficient to account for typical
ice concentrations in clouds under strong uplift conditions, we first averaged the high FBAP concentrations for the lowest, warmest aircraft samples (those with T>279K). This mean concentration of about 69 L$^{-1}$ (at 281K) was converted to INP number concentration using the Tobo et al. parameterization, assuming the parcel was lifted to higher altitudes (colder temperatures) without particle losses, but allowing for reduction in air density at higher altitudes. The predicted INP concentrations for this hypothetical case are shown as light green circles in 8b, and are
quite close to expected concentrations of primary ice based on Cooper (1986).

This approach is imperfect for a number of reasons. First, the parameterization of Tobo et al. (2013), originating at the BEACHON forested ground site, may not adequately represent the relationship between FBAP and INPs in the free troposphere over grass and cropland, which represents most of the data used here. Garcia et al. (2012) found that significant numbers of INP, which they inferred to be dominated by biogenic (heat-labile) INP, are
present in the boundary layer in this region in the autumn season and that biological INP are directly enhanced during harvesting operations. Also, the WIBS method detects only fluorescent particles larger than 0.8 μm, which may not include all biological INP. While Mason et al. (2015) present evidence that the majority of continental INP are in this size range, smaller biological particles have been detected in the free troposphere (Pratt and Prather, 2010) and as discussed earlier, some submicron organic particles (or suspendible components of larger particles
from arable soils, plants and sea spray), are known to nucleate ice (Vali, 1966; Pummer et al., 2012; Tobo et al., 2014; Wilson et al., 2015). These particles may accompany releases of FBAP, but their atmospheric inputs are presently poorly quantified or validated. Simultaneous measurements of biological and ice nucleating particles in the free troposphere would be very useful to help better understand these relationships.

Despite these limitations, this simple analysis suggests that primary fluorescent biological particles are
likely important for ice formation in mixed-phase clouds under certain conditions, in particular when lifting is strong enough to bring them to sub-zero temperatures and to counteract natural losses through sedimentation. This may occur frequently in direct association with deep convection (Phillips et al., 2009; DeLeon-Rodriguez et al., 2013) where vertical velocities are strong. In areas impacted by long-range transport (Pratt et al., 2009) or orographic and frontal uplift, it may be more sporadic. For example, biological particles were found on only two of five flights



targeting orographic clouds over the Rocky Mountains (Phillips et al., 2012). These same convective processes are expected to impact transport of mineral and other soil-dust, and dust and biological particles may be co-located (Pratt et al., 2009; Creamean et al., 2013). The greater abundance of mineral dust globally means that it will usually dominate the INP population at colder temperatures where it effectively nucleates ice, and probably explains the dominance of mineral dust in cirrus clouds (Cziczo et al., 2013). Extensive laboratory data indicate, however, that

biological particles remain the most likely source of INP in the atmosphere for clouds with temperatures warmer than ~258K. The variable and often low abundance of these INP, however, may explain why clouds sometimes remain supercooled in the atmosphere (Kanitz et al., 2011).

## 5 Conclusions and discussion

The first vertical profiles of fluorescent biological particles in the free troposphere have been presented for the

autumnal U. S. Great Plains region during daytime hours. Concentrations of FBAP larger than 0.8 µm were ~10-100 L$^{-1}$ at temperatures warmer than 270K in the atmospheric boundary layer. In the mid to upper free troposphere at temperatures less than about 255K, FBAP concentrations were usually between 0-1L$^{-1}$. Variable and sometimes high concentrations of biological particles were measured in the ~2-5 km altitude range. In this region, this altitude range coincides with temperatures where mineral dusts are less active as INPs, so INPs of biological origin are likely

critical to mixed-phase cloud formation. These data are consistent with and provide a bridge between prior measurements of biological particles near the surface, in the middle troposphere in mixed-phase clouds and in the upper troposphere in cirrus clouds.

Lower and upper bounds of FBAP concentrations were developed from the WIBS-4A data depending on fluorescent response, and translated to about an order of magnitude difference in number concentration for each

sampling period. Further studies characterizing the WIBS response to different types of biological particles in their atmospheric form are needed. In particular, preliminary analysis via scanning electron microscopy indicates that many irregular carbonaceous particle types, apparently plant-derived, were present in the IDEAS sampling region (James Anderson, personal communication). Laboratory evidence indicates that plant material such as cellulose and pectin can nucleate ice as efficiently as mineral dust (Hiranuma et al., 2015b; Möhler et al., 2016).

Filter measurements of ice nucleating particles at two altitudes over the forested BEACHON ground site showed that INP concentrations decrease with height, in accordance with the profiles of FBAP from the WIBS instrument. A simple, temperature-dependent relationship between FBAP concentration and ice nucleating particles developed for that site was used to estimate INP concentrations for the data set as a whole in the temperature range of 243-270K. Predicted INP concentrations measured in clear-air were sometimes sufficient to explain expected

concentrations of primary ice crystals for clouds in the region, but were often much lower. Several reasons were proposed for this. These include the possibility that the ground-based parameterization may not be representative of the wider data set, the potential for sub-micron biological particles to be INPs, and different atmospheric stabilities between the clear-air cases measured and cloudy conditions. If boundary-layer FBAP concentrations are hypothetically lifted to colder temperatures without losses, then predicted INP concentrations are quite close to



expected ice crystal concentrations. In-cloud measurements of FBAP and comparison with ice and droplet concentrations will be the focus of subsequent work.

    A global chemistry-climate model was employed to simulate the emissions, transport, and removal from the atmosphere of three types of biological particles: bacteria, fungal spores, and pollen. The simulated particle concentrations declined with altitude in a manner generally consistent with observations. However, the model

usually underpredicted the observed FBAP concentrations (typically by about an order of magnitude), which potentially could be due to errors in either modeled particle emissions, transport, or removal processes. A previous effort to model airborne FBAP observations by Perring et al. (2015) also found that the model consistently underpredicted observed concentrations. One likely factor in these discrepancies is that the model may be missing particle types, such as leaf litter or arable soils, which may contribute to FBAP concentrations. Another factor is that

the model does not account for the potential for agricultural activities such as harvesting to produce substantially larger emissions of biological particles, an aspect that could be relevant in this field campaign, as most flights were largely over cropland during late September and October.

    Because of the strong spatial and seasonal variability of biological particle emissions, the concentrations and profiles presented here are only representative of daytime hours in this region and season. However, the general

decrease in FBAP with height, and the sporadic incursions of high concentrations to the mid-troposphere, are likely widespread phenomena. Since ice nucleation is such a strong function of temperature, the importance of these features to ice formation will vary with time and location, but in potentially complex ways. For example, the mid-troposphere is expected to be colder at night, in the wintertime, and at high latitudes. However, these are also conditions in which biological particle loadings may be lower, due to limited convection and fewer biological

emissions. Conversely, temperatures may relatively warm in the mid-troposphere during the day, in the summer and at lower latitudes, but these are also the conditions that usually support enhanced convection and increased biological emissions. In addition, increasing temperatures and changing biospheres due to climate change are expected to modify three factors that affect the importance of biological particles on ice formation: 1) emissions and types of primary biological particles, 2) atmospheric temperature profiles, and 3) convective intensity and frequency.

*Author Contributions*

CHT oversaw the deployment and operation of instruments on the aircraft, led the data analysis and prepared the manuscript, with input from all authors. GRM, DWT and CSM modified and installed equipment and helped with data analysis and interpretation. PJD, TCJH and CSM provided INP filter samples and analysis and interpretation of them. GRK set up, ran and compiled the computational fluid dynamics studies of the inlet. SMB and MT setup, ran

and interpreted the global modeling studies. DNK calculated the seasonally averaged aerosol extinction profiles from the ARM lidar data.

*Data Availability*

Data used to create Fig. 1 were extracted from www.archive.arm.gov (SGP site Balloon-Borne Sound System, Micropulse Lidar and Multifilter Rotating Shadowband Radiometer data for 2007-2010), and the seasonal extinction

and temperature profiles are included in the first two files in the supplement (S1 and S2). Flight track data (Fig. 2)





are available in netcdf format at data.eol.ucar.edu/codiac/dss/id=378.010. FBAP and total particle concentrations, temperature, altitude and interpolated model concentrations used in Figs. 3a and 4, 5, 6, 7, and 8 are provided in Supplement S3. INP filter data for Figs. 3b and 3c are available in S4. Grid-specific model output displayed in Fig. 7 (orange diamonds) is provided in S5-S9 (each for a different flight). All supplemental data are in tab-delimited
column format with variable abbreviations and units given in the headers.

*Acknowledgements*

This material was based on work supported by the U. S. National Science Foundation under award numbers AGS-1408028 (C. Twohy),  AGS-1358495 and AGS-1036028 (P. DeMott and T. Hill) and AGS-1104642 (D. Toohey). G. Kulkarni  and S. Burrows were supported by the U.S. Department of Energy, Office of Science, BER program, at
Pacific Northwest National Laboratory (PNNL). PNNL is operated by the U.S. DOE by Battelle Memorial Institute under contract DE-AC05-76RL0 1830. James Anderson of Arizona State analyzed and provided preliminary interpretation of particle types via SEM. Greg Kok and Gary Granger of Droplet Measurement Technologies helped with modifications to the WIBS-4A. We thank Errol Korn, Gordon Maclean and Kyle Holden for technical expertise, Jeff Stith for organizing the IDEAS field program and the rest of the Research Aviation Facility staff for
implementing it so skillfully. Frank Drewnick of the Max Planck Institute for Chemistry suggested changes to the Particle Loss Calculator program for airborne operations and Yiannis Proestos of The Cyprus Institute helped with model setup. We also acknowledge the U.S. Department of Energy's Atmospheric Radiation Measurement (ARM) program and the scientists involved in providing the Southern Great Plains site MFRSR and SONDE data used in Fig. 1.

**Appendix A: CVI Inlet Subisokinetic Transport Efficiency**

When sampling by aircraft through an inlet, the transport efficiency of aerosol particles includes the inlet aspiration efficiency, the inlet transmission efficiency, and the transmission of particles downstream through the tubing between the inlet and the sampling instrument. For the IDEAS measurements, the first two efficiencies were calculated using a complete computational fluid dynamics (CFD) model with commercial software. CFD has been
used successfully with other airborne inlets (Moharreri et al., 2013) including CVIs (Laucks and Twohy, 1998; Kulkarni and Twohy, 2011). The tubing transmission efficiencies were calculated using a free IGOR "Particle Loss Calculator" program created at the Max-Planck Institute for Chemistry (von der Weiden et al., 2009). It is based on well-documented particle sampling and loss mechanisms found in the literature and has been validated for several complex tubing systems similar to what might be found in a complex aircraft-to-instrument transport environment.

**A1 Computational Fluid Dynamics modeling of CVI inlet aspiration and transmission efficiency**

A CFD model was used to calculate airflow and particle trajectories inside the airborne counterflow virtual impactor (CVI) inlet. As discussed in the main paper, the NCAR CVI was used without counterflow, as a subisokinetic inlet, to enhance the concentration of large particles during clear-air sampling. The mesh representing the CVI geometry was generated using commercial GAMBIT software (version 2.4.6) for the CFD solver. For flow regions with large





gradients in velocity and pressure magnitudes, a high density mesh was implemented. To optimize the overall mesh density, both structured and unstructured mesh elements were used, and mesh density was further refined until a grid-independent CFD solution was obtained. Approximately 350,000 cells were used to model the exterior freestream domain, the CVI inlet geometry, which included the CVI tip, porous internal tube, and solid internal tubing up to where it meets the aircraft fuselage (Fig A1). The exterior domain representing the freestream flow

around the inlet started 0.05 m upstream of the CVI inlet tip, and extended 0.05 and 0.1 m in the vertical and horizontal, respectively.

Various types of boundary conditions were used to set up the model runs. The freestream domain inlet was defined as a velocity-inlet based on the aircraft true airspeed, and the freestream flow outlet was defined as a pressure outlet. A mass-flow boundary condition was used at the downstream end of the CVI, based on the known

sample flow rate where it connects to the sample tubing inside the aircraft. The model simulations were calculated using the commercial CFD software FLUENT (ANSYS version 14.5), and the various CFD model parameters used in our study were as follows.

**Table A1: Summary of CFD model settings**

| Input summary | Settings |
|---|---|
| Models | 3D, Steady, RNG k-epsilon turbulence model, enhanced wall treatment |
| Pressure Calculation | Standard |
| Momentum, Turbulent | Second order upwind |
| Energy | First order upwind |
| Pressure-Velocity Coupling | SIMPLE |

We implemented the RNG k-ε turbulence model to consider the effect of flow turbulence, and used a Lagrangian discrete-phase model to simulate particle trajectories. Two sets of flight conditions were modeled, one representing relatively low-airspeed, low-altitude sampling conditions (freestream velocity 128 m s$^{-1}$, 670 mb ambient pressure) and one representing high-airspeed, high-altitude conditions (freestream velocity 220 m s$^{-1}$, 300 mb ambient pressure). The inlet aspiration efficiency of particles of different sizes was calculated by multiplying a)

the ratio of the inlet-plane area at the upstream domain through which particles are drawn into the CVI to the cross-sectional area of the CVI tip and b) the ratio of the freestream inlet flow velocity to the CVI tip sampling velocity. Given that the particle density and aerodynamic diameter of biological particles varies and can be less than or greater than 1.0 g cm$^{-3}$ (Després et al., 2012; Hussein et al., 2013), particles were assumed to be unit-density spheres. Size-dependent transmission efficiencies inside the CVI were calculated by assuming that any particles

whose trajectories contacted interior surfaces were not transmitted further.

**A2 Calculation of tubing transmission efficiency and net transport efficiency**

The Particle Loss Calculator (PLC) software (von der Weiden et al., 2009) calculates net particle transport through a series of tubing of different sizes, flow rates, and angles of curvature and inclination. Loss mechanisms include aerosol diffusion, sedimentation, and turbulent and inertial deposition. Thus, the program is useful for a variety of



particle sizes and sampling conditions and for complex tubing arrangements that would be difficult to model using CFD. We utilized the PLC to calculate transport of unit-density particles from where the CVI inlet ends at the aircraft fuselage, through 7.8 cm and 4.8 cm inside diameter tubing to the WIBS instrument and the ice nucleation filters. Because the default conditions of the PLC were set for ground-based conditions, air density, viscosity and mean free path values were changed to reflect the actual CVI sampling conditions for the low-speed and high-speed

cases.

     Once aspiration efficiency, inlet efficiency, and downstream tubing efficiency were determined as a function of size for both airspeed conditions, they were multiplied together to produce a net transport efficiency for nine different particle size ranges corresponding to WIBS channel diameters between 0.8 -12 μm. Due to combined effects of subisokinetic enhancement and tubing losses, transport efficiency for different sizes ranged from about 1.5

to 8 at the lower airspeeds to 2 to 12 at the high airspeeds. A linear relationship between transport efficiency and airspeed was developed for each particle size. These were used to calculate corrected number concentrations at each WIBS size channel, which were then integrated to obtain total FBAP concentrations for each 200-s interval presented in the main text. Uncertainty of the transport efficiency is discussed in Sect. 2.3 of the main text.

     For the filter samples taken on the aircraft, the size distribution of the actual ice nucleating particles was

unknown. Therefore, we assumed that the number-mean INP size was approximately 2.5 μm diameter, as estimated from ground-based BEACHON data (Huffman et al., 2013) and from wider geographical measurements by Mason et al. (2015). The airborne filter INP concentrations were corrected by a single transport efficiency corresponding to the 2.5 μm particle size at the corresponding airspeed. It should be noted, however, that INP sizes at higher altitudes may be smaller (e.g., (DeMott et al., 2010)). If the mean INP diameter were instead 1.0 μm, transport efficiencies

would be about 50% of those at 2.5 μm, leading to an increase in the INP concentrations for filters 6A and 6B (Fig. 3b and 3c) by a factor of two.

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



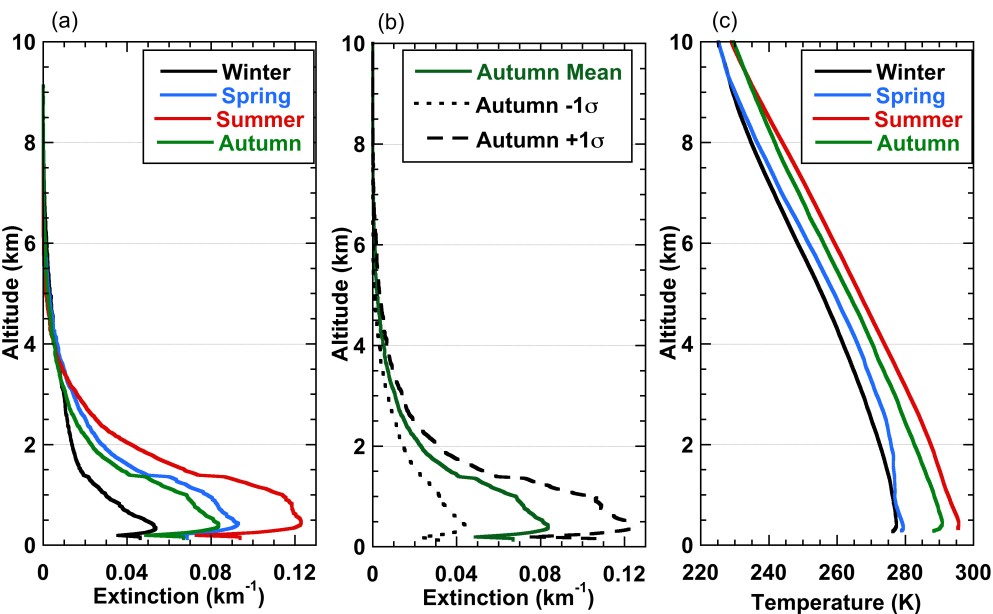

**Figure 1. DOE ARM Southern Great Plains site a) aerosol extinction coefficients as a function of altitude (km) and season for non-cloudy days. Aerosol profiles were derived using multifilter rotating shadowband radiometer (MFRSR) aerosol optical depth data (Harrison et al., 1994) as input. Since these are only directly available during daytime, in order to calculate seasonal averages, night-time values are assumed using a linear interpolation between the late afternoon and following early morning calculated lidar values. b) Aerosol extinction coefficient and corresponding 1-σ standard deviation for the autumn season. 1-σ standard deviations are given in broken lines. c) Temperature profiles from the ARM Balloon-Borne Sound System (SONDE) observations (Holdridge et al., 2011) as a function of altitude for non-cloudy days.**





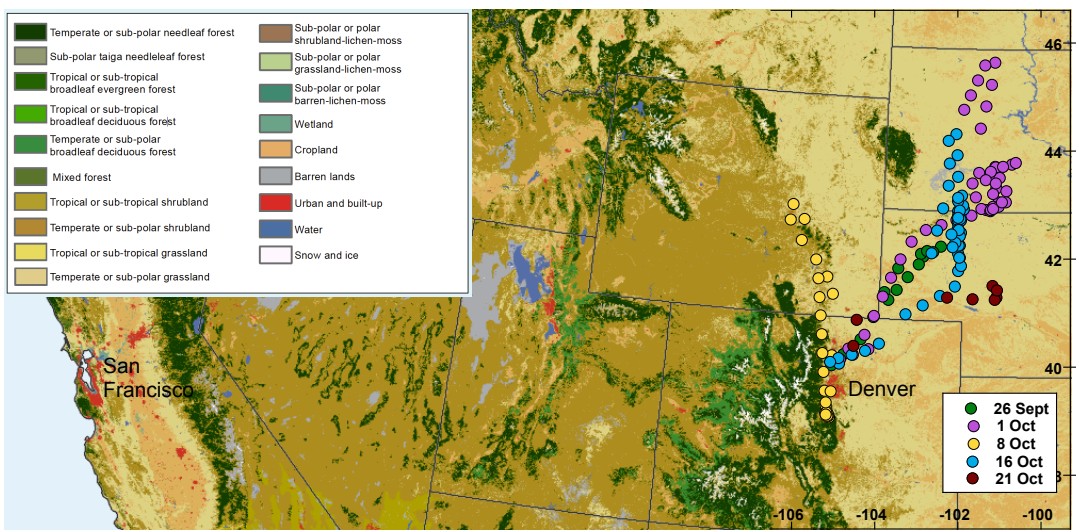

**Figure 2. Location of clear-air sampling on five flights during the IDEAS-2013 field program. Each colored dot on the right shows the location of 200 s averaged WIBS data for the dates shown in 2013. Map colors show the 2010 North American Land Cover at 250 m spatial resolution for sampling and upstream regions. Produced by Natural Resources Canada/ The Canada Centre for Mapping and Earth Observation (NRCan/CCMEO), United States Geological Survey (USGS);** *Insituto Nacional de Estadística y Geografía* **(INEGI),** *Comisión Nacional para el Conocimiento y Uso de la Biodiversidad* **(CONABIO) and** *Comisión Nacional Forestal* **(CONAFOR).**





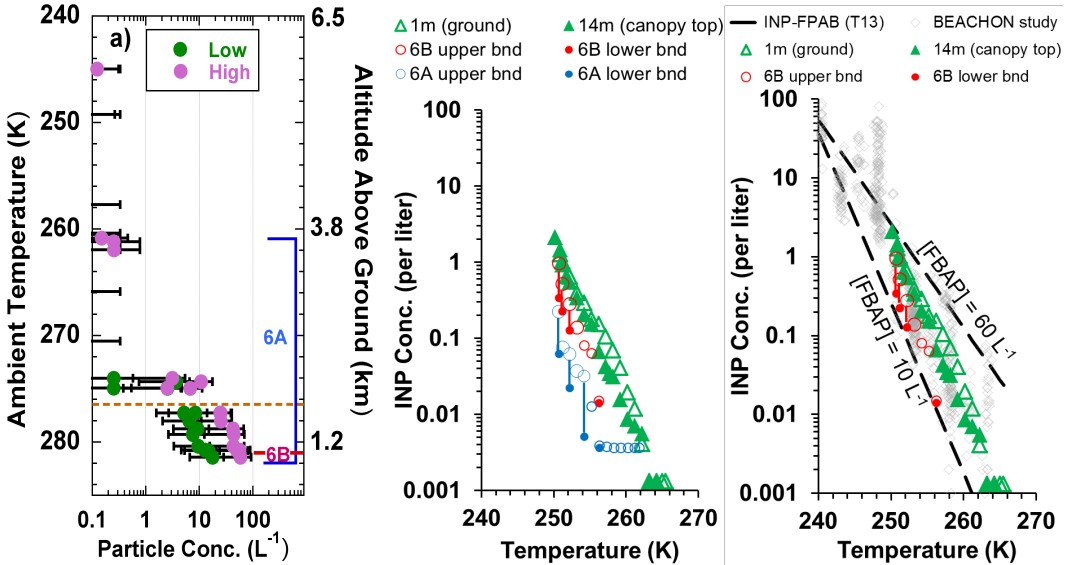

**Figure 3. a)** FBAP concentration profile over the BEACHON experimental site on 8 Oct 2013 as described in the text.
Green and magenta circles are for low and high FBAP values, respectively, based on different definitions of which
fluorescent particles are biological as discussed in the text. Top of the temperature inversion is marked with the dashed
orange line. **6A** is the location of the sample filter taken on the descent to 3638 m to 897 m above the ground at the
forested site, while **6B** is the filter taken at 1067 m. Error bars represent root-sum-square uncertainty as described in the
text. **b)** Filter-based INP spectra at 1m, 14 m, and for aircraft filters **6A** and **6B**. Upper and lower bounds are placed on
the aircraft INP samples as discussed in the manuscript, and vertical lines connect common upper and lower bounds.
Large data points in blue and red indicate data that passed a test for significance in comparison to filter blank INP
numbers. **c)** INP data for lower level samples superimposed on previous data (grey diamonds) from a continuous flow
diffusion chamber at the experimental site (Tobo et al., 2013). The two dashed lines are based on the INP vs FBAP
parameterization of Tobo et al (T13), for the range of FBAP measured by the WIBS aboard the aircraft at the lower
altitudes.





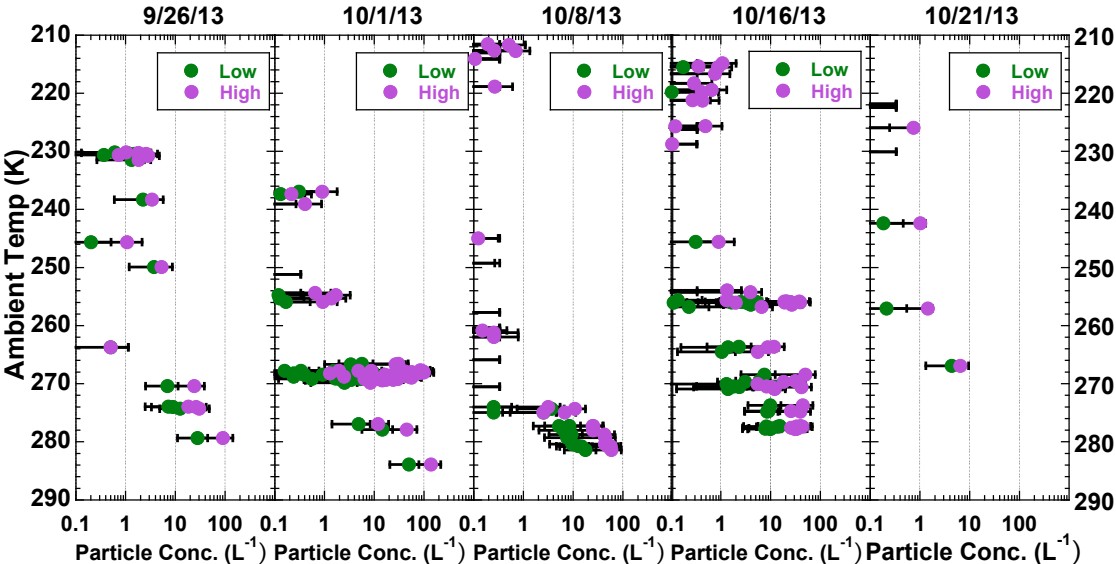

**Figure 4. Clear air temperature profiles of FBAP concentrations for each flight. Green and magenta circles are for low and high FBAP values, respectively, based on different definitions of which fluorescent particles are biological as discussed in the text. The top of the boundary layer inversion varied with flight, but was typically at temperatures warmer than 275K.**





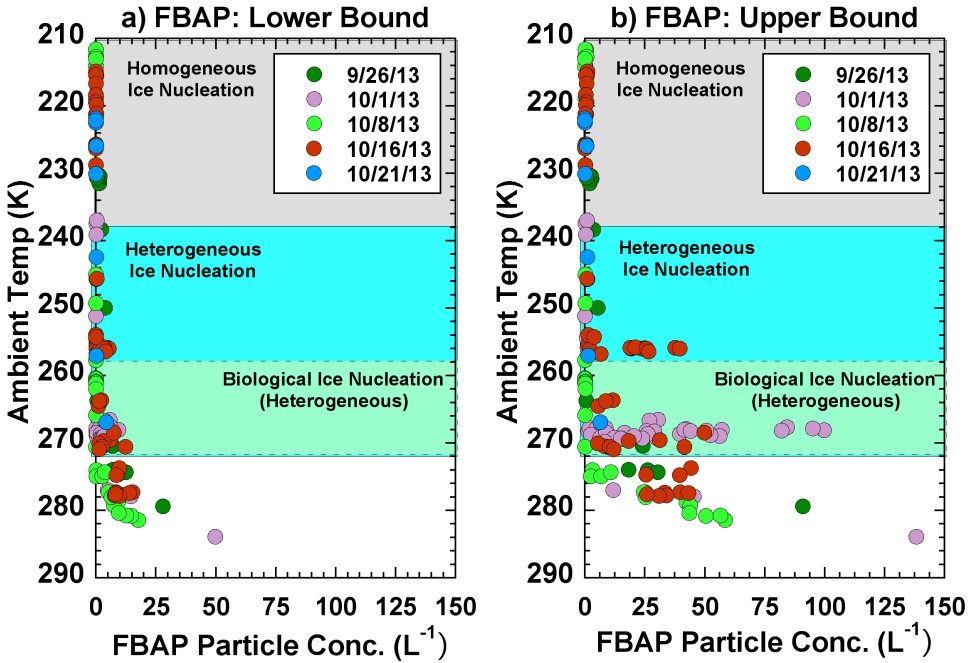


**Figure 5.** Vertical profiles of fluorescent biological particle concentration measured by the WIBS for five flights with dates and colors shown, with approximate ice nucleation temperature ranges as discussed in the text highlighted. a) lower bound values based on WIBS categorization, b) upper bound values.



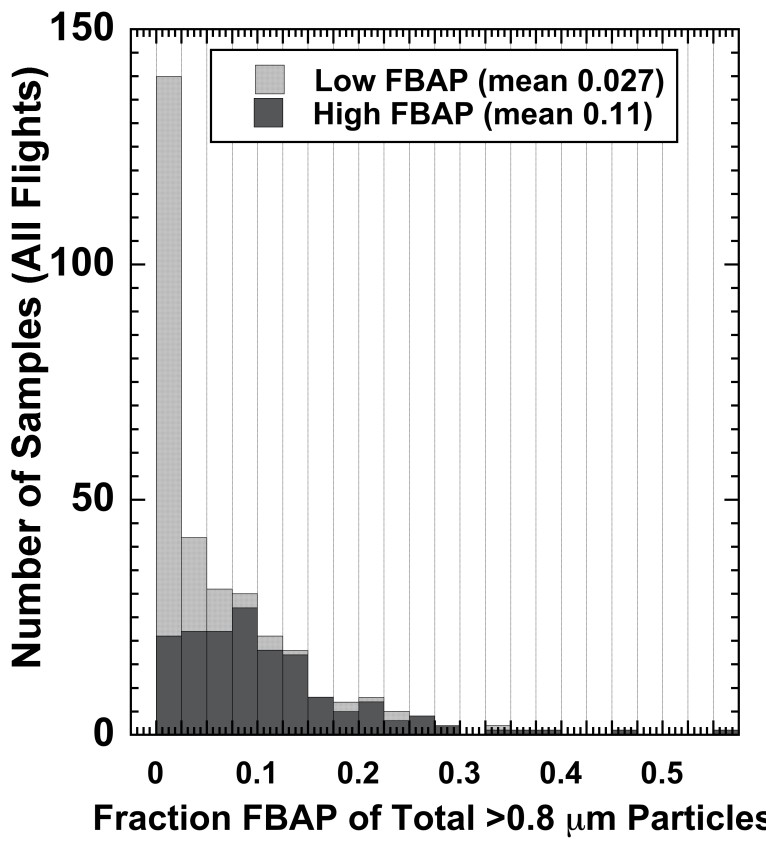


**Figure 6. Ratio of FBAP particles larger than 0.8 μm to total particle number concentration in the same size range, using expected low and high bounds as discussed in the text. Histograms of 200 second average values for WIBS samples on all five flights are plotted. Legend shows the mean fraction for each FBAP category.**




**Figure 7. Model predicted biological particle concentration (L⁻¹) as a function of altitude above ground for the five flights. Orange diamonds are all points in 25 grid boxes within the IDEAS domain (latitude range 37.30-46.63N, longitude 107.8125-98.4375 W) for 1800 UTC on each day. Black diamonds are model data interpolated to altitude, horizontal location, and time of the measurements. Green and magenta circles are the low and high FBAP values from the WIBS data, as described earlier and shown in Figs. 4 and 5. Zero values are set to 0.001 so they are visible on the log plot.**



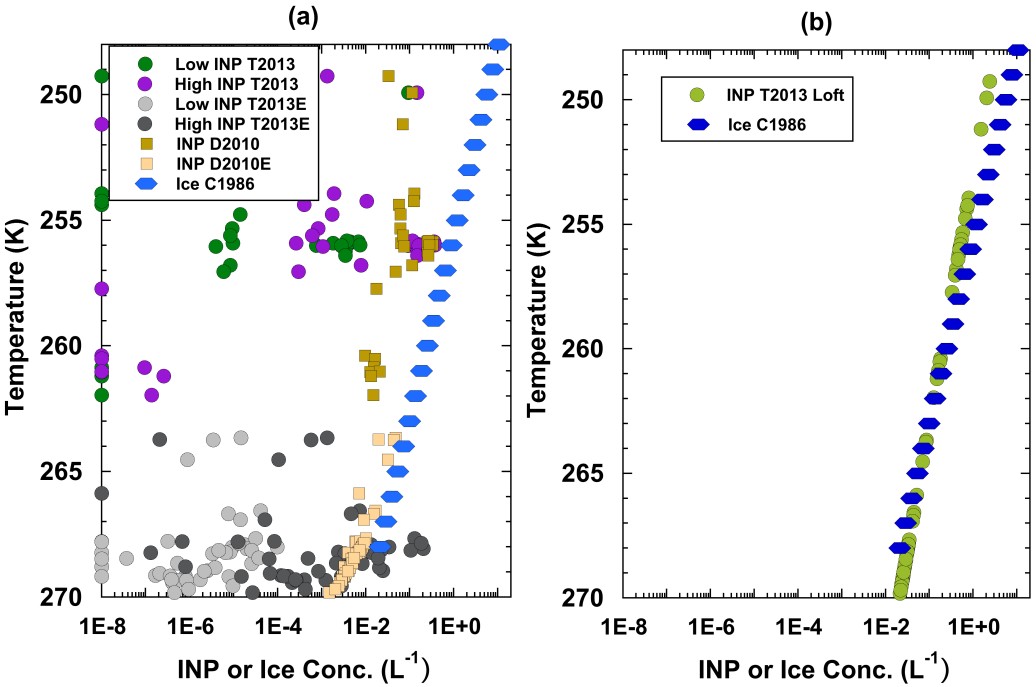

**Figure 8. (a)** Predicted INP concentration based on Tobo et al. (2013), T2013, Eqn. 3 and actual fluorescent biological particle concentrations >0.8 μm measured from the aircraft. Dark green and magenta circles are for low and high FBAP values, respectively, based on different definitions of which fluorescent particles are biological as discussed in the text. Light and dark grey circles (T2013E) are extrapolated to warmer temperatures than the data used in T2013. Zeros are plotted as $1E^{-8}$ on the log scale. Brown squares are based on the global parameterization of DeMott et al. (2010), D2010, with WIBS particle concentration >0.8 μm substituted for particles > 0.5 μm. Lighter brown squares (D2010E) are extrapolations to warmer temperatures. Dark blue hexagons represent typical concentrations of primary ice at varying temperatures in clouds as measured and parameterized by Cooper (1986), C1986. **(b)** Light green circles are a hypothetical prediction based on an FBAP concentration of 69 $L^{-1}$ at 281K, assuming boundary layer FBAP particles were lifted to colder temperatures without losses in concentration. Dark blue hexagons are ice concentrations as in (a), based on Cooper (1986). Note that all data are presented here in ambient concentrations, but as an intermediate step, FBAP and INP were converted to standard concentrations to use with the T2013 and D2010 parameterizations, as required for the equations developed in those papers. For adjusting air density, a best-fit relationship between temperature, T (K), and pressure, P (mb), was developed for all flights: $P=1428-15.753*T+0.046654*T^2$.





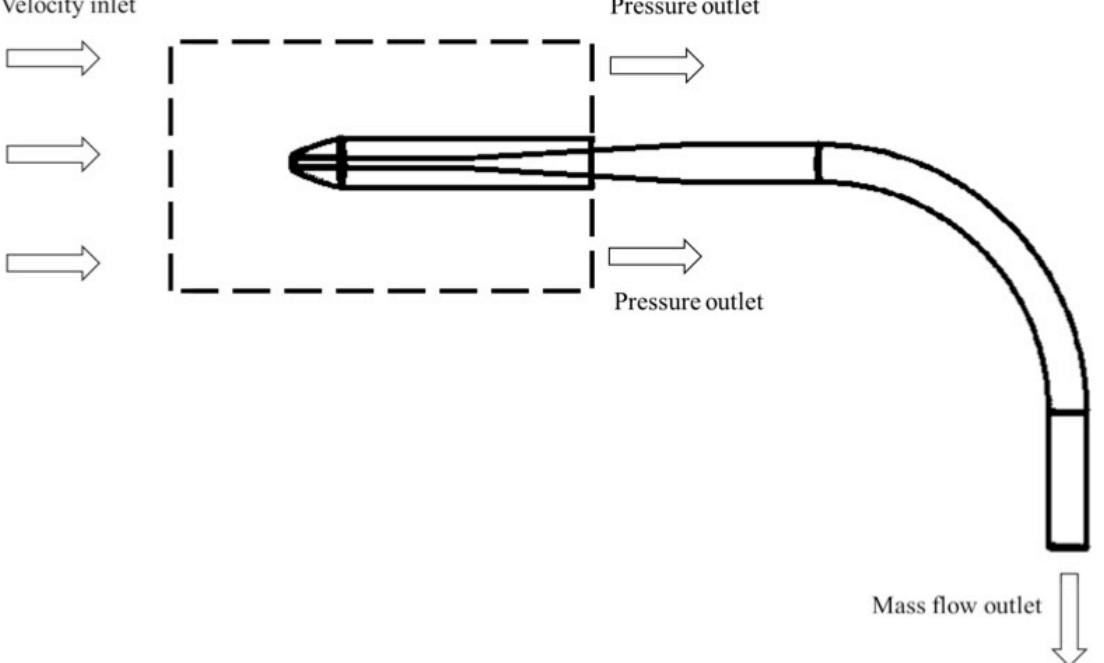

**Figure A1. Model domain and boundary conditions for the computational fluid dynamic modeling of the CVI inlet, used as a subisokinetic inlet without counterflow.**