# Peer review of "Abundance of fluorescent biological aerosol particles at temperatures conducive to the formation of mixed-phase and cirrus clouds"

_Atmospheric Chemistry and Physics, 2016_

## Referee Comment (RC1) · Anonymous Referee #1 · 12 May 2016

Reviewer Comments: "Abundance of fluorescent biological aerosol particles at temperatures conducive to the formation of mixed-phase and cirrus clouds" by C. H. Twohy et al.

Summary and general comments

In this manuscript, the vertical distribution of biological particles and ice nucleating particle (INP) concentrations over the U. S. western plains in autumn are discussed. Measurements from the boundary layer and from the free troposphere were compared. A decrease in the concentration of fluorescent biological aerosol particles (FBAP) with

height was observed with the largest variations occurring in the temperature regime of mixed-phase clouds. The vertical distribution of INPs based on the observed FBAP concentrations was derived using existing parameterizations. In addition, FBAP concentrations were compared to model results of different bioaerosol particles using the global chemistry-climate model EMAC in the sample domain.

The authors address the interest in biological aerosol particles as INPs in the atmosphere by measuring their occurrence at temperatures (altitudes) which are relevant for ice nucleation and include a comparison to atmospheric parameterizations and model results. To my knowledge, similar measurements, in particular on the vertical distribution of FBAP, have not been reported so far. The content of this paper is timely and contributes to the understanding of biological aerosol particles as INPs in the atmosphere. The manuscript is suitable for publication and its content fits well in the context of Atmospheric Chemistry and Physics. Specific reviewer comments to be addressed are given in the following.

Specific comments

Line 78-89: This paragraph is rather detailed. The reader has the impression that the properties of the field site are presented. Thus, I would rather move this paragraph to section 2 (properties of the field site) or else adapt by reducing the details in the general introduction.

Line 106: I suggest to mention earlier literature for small ice nucleation active macromolecules (INMs) from pollen rather than the given reference. Please see Pummer et al., 2012, ACP, http://www.atmos-chem-phys.net/12/2541/2012/ as an appropriate reference.

Line 115: Can the WIBS instrument be referred to as "new" device? This type of instrument has been used in a number of campaigns with published data (e.g. Toprak et al., 2013, ACP, http://www.atmos-chem-phys.net/13/225/2013/ )

Line 163: Please include if your statement includes also e.g. mineral dust particles coated with biological material. Wouldn't they fluoresce, too?

Line 326ff: It is not intuitive why the blank samples are treated like they are. For two tests of blank filters rather the average of the blanks should be subtracted (including the variability by presenting error bars).

Line 336: Please add a statement on the decrease of total aerosol number concentration. A lower INP concentration is also expected with a general decrease in total aerosol number with height and the resulting change of the aerosol size distribution.

Line 341ff: Data from this study is compared to Tobo et al. (2013). Please make clear whether the conservative or liberal approach was used by Tobo et al. (2013) when comparing to this study.

Line 345/346: "INP concentrations estimated from the WIBS data are shown" is misleading. I suggest to rephrase this part along the lines: ". . .the INP concentration derived with the parameterization by Tobo et al. (2013) and the estimated FBAP concentration are shown. . ."

Line 383: The high variability of the data is not visible "at any given temperature". Consider to change and indicate that this is particularly visible in Figure 5b and the warm temperature regime.

Line 408: The authors refer to a "variability in concentration at the same altitude throughout the region" in the model data. It is not clear what is meant by "region". Please state if it is referred to different days (panels) and rephrase accordingly. Each of the flight shows that the variability in the modeled data is not always on the same altitude throughout the measurements days.

Line 423ff: This sentence is misleading. "Large biological particles" would include e.g. pollen which are of diameters of 20 $\mu$m or larger. Earlier in the paper it has been discussed that only a small fraction of these large particles reach high altitudes relevant

for ice nucleation. Specify the size by replacing "large" by the WIBS size thresholds used.

Line 456ff: The authors state that the extrapolated data have "no basis in existing measurements". I recommend to remove these data as the scientific basis for these data points is not given.

Technical comments and language

Line 96: Replace "Fluorescent biological particle" by "FBAP"

Line 116: Hyphenation in "real time" missing. Please add to be consistent within the manuscript.

Line 120: No need in introducing FBAP again here.

Line 128: Replace "portion" by "region".

Line 229: Missing "cm" in unit "MâĎȩ cm" for resistivity.

Line 335: Please add "(6B)" after "boundary layer filter" to make it more clear which sample location you are referring to.

Line 343: Remove "WIBS" before "FBAP" for consistency.

Line 381: Replace "microbial" with "biological".

Line 383: Delete "particle" in "FBAP particle concentration" for consistency.

Line 457: Replace "this" by the specific temperature range referred to.

Line 461: "Expected" concentration is unclear. I recommend to rephrase with something along the lines: "...are well below concentrations derived with the parameterization for primary ice in clouds."

Line 485: Please be more quantitative what the term "quite close" means.

General technical remarks: Check text and figures for consistency in naming

(FBAP/FBAP particles, WIBS/WIBS-4A, ALT/Altitude Above Ground, Particle Concentration/Concentration).

Figures

Figure 3:

- Consider splitting in two figures for better readability (3a, 3b+c).

- Please label panels with a), b) and c) according to the figure caption.

- 3c: Typo in legend: It should read "INP-FBAP" instead of INP-FPAB".

- 3c: The grey data points are hardly visible (both on the screen and on print-out). Please re-color.

- 3c: Dashed lines: Please indicate in the legend what the difference makes in the two lines (low and high FBAP measurements)

- Figure caption (line 917): "." missing after "Tobo et al".

Figure 4, caption: Hyphenation of "clear-air" is missing – please add for consistency.

Figure 5: X-axis label is not consistent with the text. Delete "particle" in "FBAP Particle Conc.".

Figure 6, caption: Delete "particles" after "FBAP".

Figure 7: The legend covers the y-axis labels in all subplots. Please change.

Figure 8: The color "magenta" appears different compared to other plots and is rather purple. Please re-color to magenta to be consistent with other figures, the figure caption and text.

[Figure]

---

## Referee Comment (RC2) · JA Huffman (Referee) · 17 May 2016

The manuscript submitted by Twohy et al. presents measurements of fluorescent aerosol and ice nucleating particles (INPs) collected from the Gulfstream-V aircraft as well as modeled interpretations of these data. The investigation of biological particles at altitudes relevant for mixed phase clouds has been performed only a few times, but this is the first manuscript to describe the application of real-time fluorescence-based detection (i.e. WIBS here) along with analysis of INPs from an aircraft and showing vertical distributions. Overall, the manuscript is an excellent contribution to the literature and is very well written. The manuscript fits well in ACP, and I anticipate that it will be well received and well cited. I have some minor points that I think may help clarify certain aspects of the text, but otherwise I recommend the manuscript be published without major alteration.

Review by J. Alex Huffman

Minor Comments:
1) The presentation of the sites of observation and the timeline of comparison with previous studies is a bit confusing at times.
   a. L82 states that Figure 1 was taken from the Southern Great Plains ARM site. I would suggest putting a mark on the map in Figure 2 to highlight this.
   b. L97 discusses data "taken near Boulder, CO". Does this refer to flight data, or ground-based measurements from the BEACHON study? If referring to the aircraft data, it is confusing, because the flight tracks extend well beyond Colorado. However, if referring to the BEACHON study, I would be much more specific.
   c. L139 mentioned the "BEACHON" study (in quotes), but does not list the full name. For this journal I would suggest listing the specific name as BEACHON-RoMBAS and spelling out the acronym, per convention. The first time it is mentioned, which I believe is at this point, I would also suggest referring to the site itself, which is called the Manitou Experimental Forest Observatory (MEFO), rather than the "BEACHON project site." This may help clarify for community members familiar with the site, but not with this specific BEACHON study. An overview of the site is presented by Ortega et al., 2014.
   d. I would also suggest pointing out that the BEACHON-RoMBAS study was in July-August 2011, whereas these flights were performed in October 2013. Because the years are sometimes not reported later, it may confuse some readers who are not already familiar with these studies.
   e. L241: Here is an example where I would add the year (2011) and consider changing to MEFO or adding that information here.
   f. L304: MEFO (or the site where BEACHON-RoMBAS was performed) is near Woodland Park, CO, but not very close to Manitou Springs, CO.
2) The discussion of WIBS data as treated relatively carefully, but I would suggest changing the wording in a few places to make the statements somewhat more conservative. For example:
   a. L162 states that "most biological particles contain amino acids and other compounds that fluoresce …". True, but I would either give more detail (as in L185), or remove 'amino acids and other' from the sentence. As written it seems half-way between a specific statement and a vague one.
   b. L164, L313, L396, L423: Each of these lines give some statement implying that biological and non-biological particles can be differentiated by the WIBS. This is a nuanced discussion, as the authors mention. However, I would suggest scaling back the wording for these sections a bit to involve the word fluorescent, or some other terminology that does not inadvertently imply more knowledge than can be defended. Even though the authors do bring this up, I think it would be best to utilize terminology along the way that will help in case a reader doesn't look carefully at the sections with these important caveats.

3) Sizing
   a. The manuscript discusses "large particles" several times, but I'm not sure they are ever rigorously defined. I think the authors use 0.8 um as the lower cut for "large" because of the WIBS. Please add this unambiguously when the term "large particles" is used first. I would also add an upper size range for this, since the WIBS, and probably the inlet, do a poor job of collecting very large particles.
   b. See L395, but many other locations as well.
   c. L423
4) End of page 5 discusses how the WIBS background signal was calculated. Was the "forced trigger" calculated as one average per flight, one average for all flights, a running average within a flight? Please clarify.
5) I would suggest adding Huffman et al. (2013) to L305 and including that droplet freezing apparatus measurements were also performed alongside CFDC measurements.
6) Figure 3: Legend is a bit confusing. The caption implies that there is a difference between large and small data points, in terms of whether they passes statistical significance tests. The sizing difference is subtle, however, and I would suggest making this easier to determine from the (i.e. shaded or not …). Also, the legend repeats information (e.g. 6B upper bnd, 6B lower bnd), but I'm not sure if this is intentional or necessary.
7) Figure 7: I would suggest making one legend and putting in a location of a sixth panel. This would reduce legend redundancy and would remove the current issue that the legends cover parts of the graph and axes labels.

Specific and technical comments or corrections:
1) L 76: Comma after "Thus"
2) L110: Does the sentence need to be parenthetical?
3) L183: Add closing parentheses.
4) L204: "calibration was verified" seems a bit strong for a 1-point measurement. Maybe "check" is a better word?
5) L278: Add space between "10hPa"
6) L288: Add space between "4um"
7) L308: Add comma after "First"

References:
Ortega et al., ACP, 14, 6345-6367, doi:10.5194/acp-14-6345-2014, 2014.

---

## Author Comment (AC1) · 6 Jun 2016

Response to Anonymous Referee #1

*We thank the reviewer for his/her thorough comments, which greatly improve the revised manuscript. Please see our responses and changes below (in italics).*

Summary and general comments
In this manuscript, the vertical distribution of biological particles and ice nucleating particle (INP) concentrations over the U. S. western plains in autumn are discussed. Measurements from the boundary layer and from the free troposphere were compared. A decrease in the concentration of fluorescent biological aerosol particles (FBAP) with height was observed with the largest variations occurring in the temperature regime of mixed-phase clouds. The vertical distribution of INPs based on the observed FBAP concentrations was derived using existing parameterizations. In addition, FBAP concentrations were compared to model results of different bioaerosol particles using the global chemistry-climate model EMAC in the sample domain.

The authors address the interest in biological aerosol particles as INPs in the atmosphere by measuring their occurrence at temperatures (altitudes) which are relevant for ice nucleation and include a comparison to atmospheric parameterizations and model results. To my knowledge, similar measurements, in particular on the vertical distribution of FBAP, have not been reported so far. The content of this paper is timely and contributes to the understanding of biological aerosol particles as INPs in the atmosphere. The manuscript is suitable for publication and its content fits well in the context of Atmospheric Chemistry and Physics. Specific reviewer comments to be addressed are given in the following.

Specific comments
Line 78-89: This paragraph is rather detailed. The reader has the impression that the properties of the field site are presented. Thus, I would rather move this paragraph to section 2 (properties of the field site) or else adapt by reducing the details in the general introduction.

*These extinction profiles were in the same region, but farther south than the aircraft sampling. We feel it's still important to include them, as they provide longer-term and seasonally-averaged information on aerosol vertical distributions over a range of altitudes. Agreed that they don't belong in the Introduction, but they aren't really a description of the field site either, so we have created a short new Section 2 entitled "Aerosol extinction profiles at the ARM Southern Great Plains site" for this information. In this new section, we also describe the location of the sites. Subsequent sections are renumbered accordingly.*

Line 106: I suggest to mention earlier literature for small ice nucleation active macro-molecules (INMs) from pollen rather than the given reference. Please see Pummer et al., 2012, ACP, http://www.atmos-chem-phys.net/12/2541/2012/ as an appropriate reference.

*Thanks, we have now included both the earlier and later reference here.*

Line 115: Can the WIBS instrument be referred to as "new" device? This type of instrument has been used in a number of campaigns with published data (e.g. Toprak et al., 2013, ACP, http://www.atmos-chem-phys.net/13/225/2013/ )

*Fair enough; it is really more of a new application of it that we are presenting, but we have replaced "new" with "fast-response".*

Line 163: Please include if your statement includes also e.g. mineral dust particles coated with biological material. Wouldn't they fluoresce, too?

*Possibly; whether they are detected by the WIBS would depend on the strength of the fluorescence, and this is an area where further research is needed. For now, we have added "Particles containing mixtures of biological and non-biological material may also be classified as FBAP if their fluorescent signal is sufficiently strong."*

Line 326ff: It is not intuitive why the blank samples are treated like they are. For two tests of blank filters rather the average of the blanks should be subtracted (including the variability by presenting error bars).

*The reviewer is correct that the intuitive approach would be to average the values and then present that result with error bars. However, since the two blanks varied significantly in their INP loadings, when this approach was taken the resulting 95% confidence interval of the averaged value was smaller than the range encompassed by the two blanks. Thus, we opted to present each separately, so as to correctly represent the potential variability in the measures.*

Line 336: Please add a statement on the decrease of total aerosol number concentration. A lower INP concentration is also expected with a general decrease in total aerosol number with height and the resulting change of the aerosol size distribution.

*True; total concentration in a variety of size ranges also decreases with height. Of particular interest relative to INP are all particles (fluorescent and not) in the WIBS size range of 0.8-12 μm, which are shown in the plot below. We have added a statement to the text that indicates that total particles in this size range also decrease in concentration with height.*

[Figure]

Line 341ff: Data from this study is compared to Tobo et al. (2013). Please make clear whether the conservative or liberal approach was used by Tobo et al. (2013) when comparing to this study.

*The Tobo et al study used a UV-APS, which utilizes a single excitation wavelength of 355 nm, similar to the WIBS-4A channel C, to measure FBAP. The UV-APS signal therefore would be approximately equivalent to the sum of four WIBS-4A categories: C, AC, BC, and ABC. Neglecting other technical differences in the instruments, this FBAP concentration should be between the conservative (using categories AC and ABC) and liberal approaches (using A, C, AB, AC, BC and ABC) in our analysis, and this is true for the MEFO site case presented. These differences between the WIBS-4A and UV-APS are now explained in the text in the second paragraph of the Results section. Later when introducing the Tobo et al. 2013 parameterization, we note that it is based on UV-APS data.*

Line 345/346: "INP concentrations estimated from the WIBS data are shown" is misleading. I suggest to rephrase this part along the lines: "... the INP concentration derived with the parameterization by Tobo et al. (2013) and the estimated FBAP concentration are shown

*We have tried to make this more clear; the entire section now reads: "Additionally, ice nucleating particle concentrations were estimated as a function of WIBS-4A FBAP concentrations measured from the aircraft, using a recent parameterization by Tobo et al. (2013) based on the concentration of FBAP >0.5 µm. Using measured low-level FBAP concentrations of 10 $L^{-1}$ to 60 $L^{-1}$ (approximate low and high values in Fig. 3a), the INP*

*concentrations derived from the parameterization by Tobo et al. (2013) are shown as the two dashed black lines in Fig. 3b."*

Line 383: The high variability of the data is not visible "at any given temperature". Consider to change and indicate that this is particularly visible in Figure 5b and the warm temperature regime.

*True; we have changed the sentence to be more specific: "Upper-bound FBAP concentrations (Fig 5b) are most variable, particularly in the ~270 K to 255 K temperature region where they are likely most important to ice formation in mixed-phase clouds."*

Line 408: The authors refer to a "variability in concentration at the same altitude throughout the region" in the model data. It is not clear what is meant by "region". Please state if it is referred to different days (panels) and rephrase accordingly. Each of the flight shows that the variability in the modeled data is not always on the same altitude throughout the measurements days.

*We meant variability at each of the 25 grid squares (on a single day) within the considered model domain, which encompasses all the IDEAS flight data. We have made this sentence more specific, and also moved it up to immediately after describing the orange diamonds so what we mean is more clear: "It is interesting to note that the model often predicts 1-2 orders of magnitude variability in biological particle concentration at the same altitude in different grid boxes throughout the sampling region (37.30-46.63 N, 107.81-98.44 W)."*

Line 423ff: This sentence is misleading. "Large biological particles" would include e.g. pollen which are of diameters of 20 μm or larger. Earlier in the paper it has been discussed that only a small fraction of these large particles reach high altitudes relevant for ice nucleation. Specify the size by replacing "large" by the WIBS size thresholds used.

*Changed to "0.8 to 12 μm FBAP". In addition, we have added a statement at the end of Section 3.3 stating that the lack of including intact pollen in the results is likely insignificant for total FBAP concentrations, since both prior measurements and the EMAC model indicate pollen number concentrations are typically order of magnitudes smaller than those for bacteria and fungal spores.*

Line 456ff: The authors state that the extrapolated data have "no basis in existing measurements". I recommend to remove these data as the scientific basis for these data points is not given.

*We feel that it's important to include the warmer temperatures that comprise a significant portion of the WIBS data in the temperature range where biological INP may be important, as long as it's clear that the data is extrapolated and so less certain. We have reworded this sentence and the one before it as follows:*
*"Our analysis used the T2013 parameterization for the temperature range of the BEACHON-*

*RoMBAS INP data set (243K-263K), plus an extrapolation to seven degrees warmer to incorporate a broader range of FBAP data and temperatures where biological INPs are potentially important. Predicted INP number concentrations as a function of ambient temperature are shown in colored circles (T2013) in Fig. 8a. Points where the parameterization was extrapolated to warmer temperatures are colored grey (T2013E) to indicate that they have greater uncertainty."*

**Technical comments and language**

Line 96: Replace "Fluorescent biological particle" by "FBAP"
*Done.*

Line 116: Hyphenation in "real time" missing. Please add to be consistent within the manuscript.
*Done.*

Line 120: No need in introducing FBAP again here.
*This is now the first occurrence of the term since we have moved the section on the extinction measurements farther down. Accordingly, that now-later, extraneous definition of FBAP has been removed.*

Line 128: Replace "portion" by "region".
*Done.*

Line 229: Missing "cm" in unit "MΩ cm" for resistivity.
*Added.*

Line 335: Please add "(6B)" after "boundary layer filter" to make it more clear which sample location you are referring to.
*Done.*

Line 343: Remove "WIBS" before "FBAP" for consistency.
*Done.*

Line 381: Replace "microbial" with "biological".
*Done.*

Line 383: Delete "particle" in "FBAP particle concentration" for consistency.
*Done.*

Line 457: Replace "this" by the specific temperature range referred to.
*We have removed "in this temperature range", since we are referring to the entire range of temperatures shown in Fig. 8.*

Line 461: "Expected" concentration is unclear. I recommend to rephrase with something along the lines: "are well below concentrations derived with the parameterization for primary ice in clouds."
*Changed, thank you.*

Line 485: Please be more quantitative what the term "quite close" means.
*Changed to "within a factor of two to three of".*

General technical remarks: Check text and figures for consistency in naming (FBAP/FBAP particles, WIBS/WIBS-4A, ALT/Altitude Above Ground, Particle Concentration/Concentration).

*We have made these consistent throughout. We've changed "FBAP particles" to "FBAP", and "WIBS" to "WIBS-4A", except for when we are discussing past results which may have used similar fluorescence-based instruments from the same dual-excitation/emission family, but without the 4A model number. On the plot legends, "Altitude", "Temperature" and "Particle Conc." have been standardized.*

**Figures**
*Thank for your careful review of the figure details.*
Figure 3: - Consider splitting in two figures for better readability (3a, 3b+c).
*Done and explanatory text adjusted.*
- Please label panels with a), b) and c) according to the figure caption.
*Done.*
- 3c: Typo in legend: It should read "INP-FBAP" instead of INP-FPAB".
*Done.*
- 3c: The grey data points are hardly visible (both on the screen and on print-out. Please re-color.
*Done; they are darker now.*
- 3c: Dashed lines: Please indicate in the legend what the difference makes in the two lines (low and high FBAP measurements)
*This is too difficult to condense into the legend, but we have explained it in more detail in the caption.*
- Figure caption (line 917): "." missing after "Tobo et al".
*Done.*
Figure 4, caption: Hyphenation of "clear-air" is missing – please add for consistency.
*Done.*
Figure 5: X-axis label is not consistent with the text. Delete "particle" in "FBAP Particle Conc.".
*We have changed to just "Particle Conc." as for the other plots, with caption explaining the particle type plotted.*
Figure 6, caption: Delete "particles" after "FBAP".
*Done.*
Figure 7: The legend covers the y-axis labels in all subplots. Please change.
*Done.*
Figure 8: The color "magenta" appears different compared to other plots and is rather purple.
*Done.*

**Other minor corrections made to manuscript by authors:**

*We have modified the following statement: "Pratt et al. (2009) and Creamean et al. (2013) reported that biological particles sometimes **dominated** ice residuals in mid-level clouds over the western United States" to: "Pratt et al. (2009) and Creamean et al. (2013) reported that biological particles sometimes **comprised a large fraction** of ice residuals in mid-level clouds over the western United States." The prior statement was somewhat misleading, since in the Pratt et al. paper, mineral dust actually dominated over biological particles.*

*We have modified the sentence immediately before the "Conclusions and discussion" section as follows: **"The variable and often low abundance of these INP, however, may explain why clouds sometimes remain supercooled in the atmosphere, particularly at warmer temperatures (Kanitz et al., 2011; Komurcu et al., 2014)."** This reflects new information and an important data set on supercooled clouds not known to us at the time of submission (added Komurcu reference).*

---

## Author Comment (AC2) · 6 Jun 2016

Response to Referee #2 (J Alex Huffman)

*We appreciate the positive comments of Professor Huffman, and thank him for his critique. Please see responses below (in italics).*

The manuscript submitted by Twohy et al. presents measurements of fluorescent aerosol and ice nucleating particles (INPs) collected from the Gulfstream-V aircraft as well as modeled interpretations of these data. The investigation of biological particles at altitudes relevant for mixed phase clouds has been performed only a few times, but this is the first manuscript to describe the application of real-time fluorescence-based detection (i.e. WIBS here) along with analysis of INPs from an aircraft and showing vertical distributions. Overall, the manuscript is an excellent contribution to the literature and is very well written. The manuscript fits well in ACP, and I anticipate that it will be well received and well cited. I have some minor points that I think may help clarify certain aspects of the text, but otherwise I recommend the manuscript be published without major alteration.

Minor Comments
1) The presentation of the sites of observation and the timeline of comparison with previous studies is a bit confusing at times.
a. L82 states that Figure 1 was taken from the Southern Great Plains ARM site. I would suggest putting a mark on the map in Figure 2 to highlight this.

*Great idea. We have extended the SE quadrant of Fig. 2 so that the ARM site is now marked on the map, and mentioned in the caption.*

b. L97 discusses data "taken near Boulder, CO". Does this refer to flight data, or ground-based measurements from the BEACHON study? If referring to the aircraft data, it is confusing, because the flight tracks extend well beyond Colorado. However, if referring to the BEACHON study, I would be much more specific.

*Boulder was near where the aircraft was based and was a convenient site to compare temperature soundings to the ARM site soundings. For clarity, we have expanded the text to: "Fluorescent biological particle measurements described later in this paper were taken farther north in Colorado, Wyoming, North Dakota and Nebraska. For comparison, seasonally-averaged surface temperatures at Boulder, Colorado are about 2K-5K colder than at the ARM site."*

c. L139 mentioned the "BEACHON" study (in quotes), but does not list the full name. For this journal I would suggest listing the specific name as BEACHON-RoMBAS and spelling out the acronym, per convention. The first time it is mentioned, which I believe is at this point, I would also suggest referring to the site itself, which is called the Manitou Experimental Forest Observatory (MEFO), rather than the "BEACHON project site." This may help clarify for community members familiar with the site, but not with this specific BEACHON study. An overview of the site is presented by Ortega et al., 2014.

*We have implemented both these suggestions, replacing the BEACHON study with "BEACHON-RoMBAS" and the BEACHON site with "MEFO" site, as appropriate throughout. We've also moved the discussion of the BEACHON-RoMBAS experiment and MEFO site down to section 3.1, with the Ortega reference, where all this information can be together.*

d. I would also suggest pointing out that the BEACHON-RoMBAS study was in July-August 2011, whereas these flights were performed in October 2013. Because the years are sometimes not reported later, it may confuse some readers who are not already familiar with these studies.

*Done in section described above and below.*

e. L241: Here is an example where I would add the year (2011) and consider changing to MEFO or adding that information here.

*Done.*

f. L304: MEFO (or the site where BEACHON-RoMBAS was performed) is near Woodland Park, CO, but not very close to Manitou Springs, CO.

*Changed to Woodland Park.*

2) The discussion of WIBS data as treated relatively carefully, but I would suggest changing the wording in a few places to make the statements somewhat more conservative. For example:
a. L162 states that "most biological particles contain amino acids and other compounds that fluoresce ...". True, but I would either give more detail (as in L185), or remove 'amino acids and other' from the sentence. As written it seems half-way between a specific statement and a vague one.

*Removed amino acids as suggested.*

b.L164, L313, L396, L423: Each of these lines give some statement implying that biological and non-biological particles can be differentiated by the WIBS. This is a nuanced discussion, as the authors mention. However, I would suggest scaling back the wording for these sections a bit to involve the word fluorescent, or some other terminology that does not inadvertently imply more knowledge than can be defended. Even though the authors do bring this up, I think it would be best to utilize terminology along the way that will help in case a reader doesn't look carefully at the sections with these important caveats.

*Understood; we have changed the first discussion in Section 2.3 to: "Therefore the WIBS-4A may be used to distinguish fluorescent particles that are predominantly biological from non-fluorescent particles that are predominantly non-biological (Pöhlker et al., 2012; Huffman et al., 2013)", and changed **biological particles** to **FBAP** or similar, more appropriate terminology throughout when referring to WIBS measurements. In one case (current line 217), we have changed biological to supermicron, which is more correct in this instance.*

3) Sizing

a. The manuscript discusses "large particles" several times, but I'm not sure they are ever rigorously defined. I think the authors use 0.8 um as the lower cut for "large" because of the WIBS. Please add this unambiguously when the term "large particles" is used first. I would also add an upper size range for this, since the WIBS, and probably the inlet, do a poor job of collecting very large particles.

*Good point, both to be rigorous in describing what we are presenting and also since other studies may use a slightly different size range. The lower limit is indeed due to WIBS detection limits and the effective upper limit is about 12 microns due to system transmission efficiencies. We have specified this size range now in the Abstract and added an explanation of these size limits in the second to last paragraph of Section 2: "Based on size-dependent concentration corrections for inlet aspiration and transmission efficiency described in Appendix A, net efficiency for particles larger than 12 μm diameter was less than 2%. Detection of fluorescent particles smaller than 0.8 in diameter is limited by the sensitivity of the WIBS detectors (Gabey et al., 2010). Therefore, when presenting measured concentrations or properties of "fluorescent biological particles" or "FBAP" in this paper, only particles between 0.8 μm and 12 μm in diameter are represented. "*

b. See L395, but many other locations as well.  c. L423

*We have now more rigorously specified the measured size range (0.8-12 μm) where appropriate throughout.*

4) End of page 5 discusses how the WIBS background signal was calculated. Was the "forced trigger" calculated as one average per flight, one average for all flights, a running average within a flight? Please clarify.

*We have inserted a more detailed explanation of how the forced triggers were used: "Forced trigger measurements were performed at the beginning and end of each flight, during which time the instrument fires the UV light sources in the absence of particles to measure the background signal. The background signal averages and standard deviations were linearly interpreted over each flight. Only fluorescent signals larger than the forced-trigger average value plus 2.5 standard deviations are included in the data presented here."*

5) I would suggest adding Huffman et al. (2013) to L305 and including that droplet freezing apparatus measurements were also performed alongside CFDC measurements.
*Now added to beginning of Section 3.1.*

6) Figure 3: Legend is a bit confusing. The caption implies that there is a difference between large and small data points, in terms of whether they passes statistical significance tests. The sizing difference is subtle, however, and I would suggest making this easier to determine from

the (i.e. shaded or not …). Also, the legend repeats information (e.g. 6B upper bnd, 6B lower bnd), but I'm not sure if this is intentional or necessary.

*We have combined your suggestions and those of the other reviewer to combine 3b and 3c into a new 3b plot, which is larger and more legible. The legends are also modified for clarity. We have retained, however, two sets of 6A and 6B symbols in the legend, since upper and lower bounds use slightly different symbols.*

7) Figure 7: I would suggest making one legend and putting in a location of a sixth panel. This would reduce legend redundancy and would remove the current issue that the legends cover parts of the graph and axes labels.
*Done as requested.*

Specific and technical comments or corrections:
1) L 76: Comma after "Thus"
*Done.*
2) L110: Does the sentence need to be parenthetical?
*Parentheses removed.*
3) L183: Add closing parentheses.
*Done.*
4) L204: "calibration was verified" seems a bit strong for a 1-point measurement. Maybe "check" is a better word?
*Yes, changed.*
5) L278: Add space between "10hPa"
*Done.*
6) L288: Add space between "4um"
*Done.*
7) L308: Add comma after "First"
*Done.*

***Other minor corrections made to manuscript by authors:***
*We have modified the following statement: "Pratt et al. (2009) and Creamean et al. (2013) reported that biological particles sometimes **dominated** ice residuals in mid-level clouds over the western United States" to: "Pratt et al. (2009) and Creamean et al. (2013) reported that biological particles sometimes **comprised a large fraction** of ice residuals in mid-level clouds over the western United States." The prior statement was somewhat misleading, since in the Pratt paper, mineral dust actually dominated over biological particles.*

*We have modified the sentence immediately before the "Conclusions and discussion" section as follows: "**The variable and often low abundance of these INP, however, may explain why clouds sometimes remain supercooled in the atmosphere, particularly at warmer temperatures (Kanitz et al., 2011; Komurcu et al., 2014).**" This reflects new information and an important data set on supercooled clouds not known to us at the time of submission (added Komurcu reference).*